# Marine toxin domoic acid alters nitrogen cycling in sediments

Zelong Li[1,3], Jing Wang [1] ✉, Hao Yue [1,3], Miaomiao Du[1], Yuan Jin[2] & Jingfeng Fan [2] ✉

As a red tide algal toxin with intense neurotoxicity distributed worldwide, domoic acid (DA) has attracted increasing concerns. In this work, the integrative analysis of metagenome and metabolome are applied to investigate the impact of DA on nitrogen cycling in coastal sediments. Here we show that DA can act as a stressor to induce the variation of nitrogen (N) cycling by altering the abundance of functional genes and electron supply. Moreover, micro-ecology theory revealed that DA can increase the role of deterministic assembly in microbial dynamic succession, resulting in the shift of niches and, ultimately, the alteration in N cycling. Notably, denitrification and Anammox, the important process for sediment N removal, are markedly limited by DA. Also, variation of N cycling implies the modification in cycles of other associated elements. Overall, DA is capable of ecosystem-level effects, which require further evaluation of its potential cascading effects.

Domoic acid (DA), also labeled as marine phantom toxin, belongs to a kind of potent neurotoxin produced by diatoms of the genus *Pseudo-nitzschia*[1,2]. As structurally similar toxins to glutamate, DA can bind directly to glutamate receptors in vivo, promoting the release of endogenous glutamate and thereby causing damage to neural tissues[3,4]. Nowadays, global warming and severe marine pollution have led to a rise in the frequency and prevalence of periodic and sudden harmful algal blooms (HABs), which makes DA pervasive in the global ocean[5–7]. As such, the physiological and ecological effects of DA toward marine environment merit critical investigation.

Despite the fact that DA is primarily produced by red-tide diatoms, it exists not only in the marine photic zone but also evidently exhibits vertical migration characteristics from the surface seawater to the seabed[8–10]. As DA-containing algal cells aggregate with other biological pellets, they can form marine snow particles that migrate rapidly to the marine aphotic zone under the action of gravity, where they release DA and algal organic matter (AOM) into the sediment[8,11]. More importantly, the marine benthic environment has been proven to be the primary destination for DA[8]. It has been confirmed by the 15-year continuous fieldwork that the DA released by the decay of *Pseudo-nitzschia* cells in the benthic environment is primarily present in the

aqueous phase, i.e., in the form of dissolved DA (dDA)[9]. The concentration of dDA is commonly about 0.6 mg l$^{-1}$, and its maximum concentration can reach 4.2 mg l$^{[-19]}$. In addition, a minute amount of DA released from algal cells will be adsorbed by the sediment (pDA), and its concentration can reach the level of ng/g wet sed[9,10].

In general, the current research on the physiological and ecological effects of DA is solely confined to animals and phytoplankton. Originally identified as the culprit for the episodes of fatal human poisoning in the late 1980s[12], DA was subsequently proven to be associated with marine animal casualties at different trophic levels through the transfer in food webs[13,14], such as copepods, whelks, sealions, sea otters, whales, etc.[7,15,16]. Toxic *Pseudo-nitzschia* blooms were also reportedly to coincide with the breeding season of marine wildlife, which could be transferred to marine mammal fetuses via utero or milk, thereby posing long-lasting adverse effect to neuron in developing fetuses[17]. Besides, growth of phytoplankton can be inhibited by DA as well[18,19], further confirming the urgent threat of this biotoxin to the marine environment.

Serving as the major engine for the biogeochemical cycles, microorganisms play integral roles in marine ecosystem[20]. As of yet, there is little information available regarding the effects of DA on

[1]Key Laboratory of Industrial Ecology and Environmental Engineering (Ministry of Education), School of Environmental Science and Technology, Dalian University of Technology, Dalian 116024, PR China. [2]Marine Ecology Department, National Marine Environmental Monitoring Center, Dalian 116023, PR China. [3]These authors contributed equally: Zelong Li, Hao Yue. ✉e-mail: jwang@dlut.edu.cn; jffan@nmemc.org.cn

microorganisms, with sporadic studies coming to the conclusion that DA can inhibit the growth of the strains isolated from marine environment[21]. Based on our previous study, trace amounts of DA (1 mg l⁻¹) can induce degraders to secrete more extracellular polymeric substances (EPS) during its biotransformation, which serves as a protective strategy against external stress[22]. Taken together, the available information implies that DA may act as a stressor toward microorganisms. Consequently, we speculate that DA is able to exert effects on microbes and, ultimately, alter the community structure and functions. Microbial functions are inextricably linked to biogeochemical cycles, and N-cycle functional microorganisms are reported to be vulnerable to some micropollutants such as microplastics[23], antibiotics[24], etc. In light of this, we therefore proposed our hypothesis that DA is capable of affecting N cycling in sediments. In this work, ¹⁵N isotope-tracing, metagenome and metabolome were integrated to test our hypothesis. To our best knowledge, this work will provide an important perspective on the ecological impact of DA in the marine environment.

## Result

### Effects of DA on potential N transformation rates

It was reported that DIN in overlying water is mainly from sediments[25], so the variation of DIN concentrations in overlying water was monitored, so as to infer the potential N transformation process in sediments. Generally, it can be observed that the overlying water DIN concentrations remain essentially stable in the CK group. The $NH_4^+$, $NO_3^-$, and $NO_2^-$ concentrations are in the ranges of 1.6–23.7, 0.46–6.4, and 0.12–0.56 nM, respectively. By contrast, the DIN concentrations

separately exhibit obvious fluctuations in AD$_{0.1}$, AD$_{0.5}$, AD$_{1.0}$ and AOM groups (Fig. 1a). To be specific, the accumulation of $NH_4^+$ can be clearly observed in the initial experiment, and then the $NH_4^+$ concentrations in groups of AD$_{0.1}$, AD$_{0.5}$, AD$_{1.0}$ and AOM reach the maximum on day 6 (15.3 nM), 15 (22.2 nM), 18 (23.7 nM) and 6 (16.0 nM), respectively. As for $NO_3^-$ and $NO_2^-$, their concentrations rise briefly and then decline rapidly in groups of AD$_{0.1}$ and AOM. While in groups of AD$_{0.5}$ and AD$_{1.0}$, notable increases in $NO_3^-$ and $NO_2^-$ concentrations can be seen at the beginning of the third day. The maximum concentrations of $NO_3^-$ and $NO_2^-$ reach 4.2 and 0.43 nM in the AD$_{0.5}$ group and 6.4 and 0.56 nM in the AD$_{1.0}$ group, respectively. Overall, the DIN concentrations in groups of AD$_{0.1}$ and AOM exhibit almost consistent variations, while compared with the CK and AOM groups, the DIN concentration in groups of AD$_{0.5}$ and AD$_{1.0}$ are higher. Therefore, we speculate that DA with concentrations of either 0.5 or 1 mg l⁻¹ can alter the N transformation process.

Furthermore, the DA concentration within the system declined gradually after day 9 in all three treatments (Fig. S1). Taking AD$_{0.5}$ as an illustrative example, it is notably observed that DIN concentrations also exhibit a declining trend as DA concentrations decrease. This observation suggests a diminishing suppression effect of DA on the sediment N removal process (e.g., denitrification and Anammox). It's worth noting that while DA concentrations start to decline from the 9th day, the decrease in DIN concentrations occurs after the 9th day. This could be attributed to the fact that although the inhibitory effect of DA on sediment N removal diminishes, the N removal efficiency remains lower than the sediment's N production efficiency (e.g., nitrification and DNRA). Only after a dynamic equilibrium is achieved

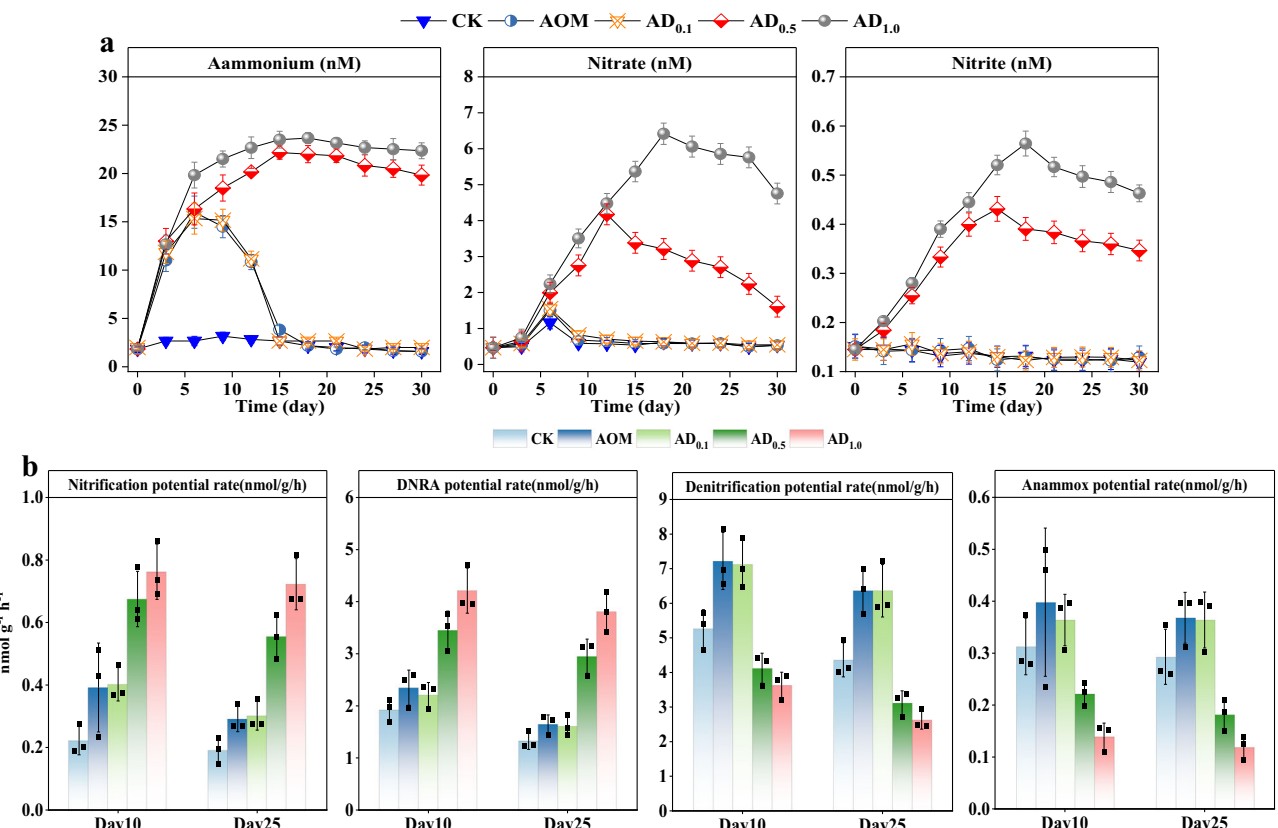

**Fig. 1 | Changes of inorganic nitrogen concentrations and potential nitrogen transformation rates in the systern.** The variations of ammonium, nitrate, and nitrite concentrations in the overlying water (**a**) (n = 3 biological replicates). The potential rates of nitrification, DNRA (dissimilating nitrate reduction to ammonium), denitrification and Anammox in sediments under different treatments (**b**) (n = 3 biological replicates). CK = without any modification, AOM = modified with algal organic matter only, AD$_{0.1}$ = modified with algal organic matter and 0.1 mg/l domoic acid, AD$_{0.5}$ = modified with algal organic matter and 0.5 mg/l domoic acid, AD$_{1.0}$ = modified with algal organic matter and 1.0 mg/l domoic acid. Error bars represent mean ± standard deviation.

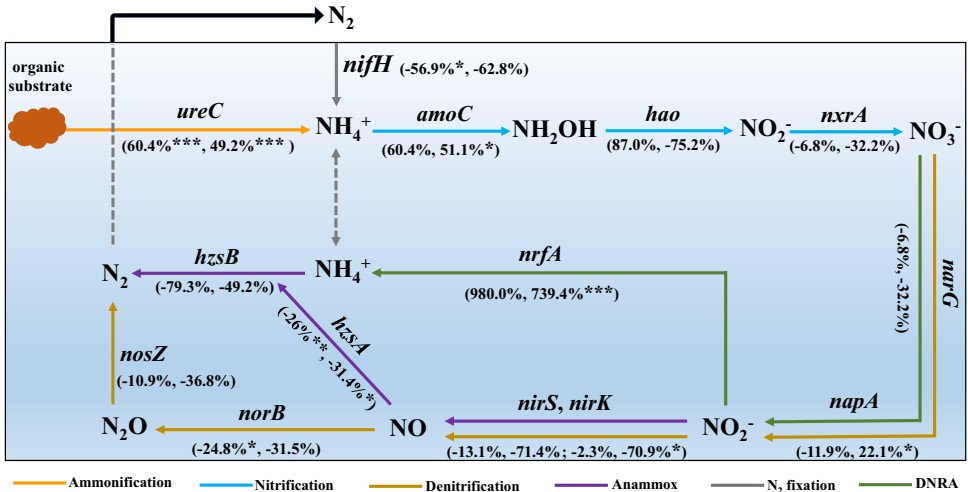

**Fig. 2 | Alterations of genes relative abundance in nitrogen cycling between AD$_{0.5}$ and CK groups.** The percentage changes of each gene in brackets represent the change in AD$_{0.5}$ group relative to the CK group on day 10 and day 25, respectively (100% × ((mean value in AD$_{0.5}$ group/mean value in CK group) − 1)). Nitrogen cycling: ammoniation, nitrification, denitrification, Anammox, N$_2$ fixation, DNRA (dissimilatory nitrate reduction to ammonium). AD$_{0.5}$ = modified with algal organic matter and 0.5 mg/l domoic acid, CK = without any modification. *$p \leq 0.05$, **$p \leq 0.01$, ***$p \leq 0.001$.

between N production and removal efficiencies, DIN concentrations gradually decrease.

To have an insight into the N transformation process in sediments, potential rates of nitrification, dissimilatory nitrate reduction to ammonium (DNRA), denitrification and Anammox were examined on day 10 and 25. The obtained data show that compared with the groups of CK and AOM, the DNRA and nitrification rates of AD$_{0.5}$ and AD$_{1.0}$ groups are substantially higher on day 10 and 25. At the same time, denitrification and Anammox, the major forms of sediment N removal, were inhibited in AD$_{0.5}$ and AD$_{1.0}$ groups (Fig. 1b). Especially, DA presents significant dose effects on sediment N transformation in the groups of AD$_{0.5}$ and AD$_{1.0}$. It should be stated that although the potential N transformation rate cannot reflect the in-situ N transformation rate, it can provide an important reference for our judgment of the effect of DA on sediment N transformation. Overall, the variation of DIN concentrations and potential N transformation rates have validated our hypothesis that DA can alter N cycling in sediment.

### Effect of DA on N metabolism

Since the concentration of DA in the marine benthic environment is close to 0.5 mg/l[9], and based on our data, DA with a concentration of 0.5 mg/l can alter the N cycling, the groups of AD$_{0.5}$, AOM, and CK were subjected to metagenomic analysis (on day 10 and 25), to comprehensively understand the mechanism of how DA affects sediment N cycling. The results demonstrate that the relative abundance of genes involved in N cycling differs in different treatments (Figs. 2 and S2 and Table S1). On the 10th and 25th days, in comparison to the AOM and CK groups, the AD$_{0.5}$ group exhibited the lowest relative abundance of genes associated with denitrification (*narG*, *nirS*, *nirK*, *norB*, *nosZ*), N$_2$ fixation (*nifH*) and Anammox (*hzsA*, *hzsB*, *nirS*, *nirK*) (Figs. 2 and S2). On the contrary, within the three treatments, the AD$_{0.5}$ group displayed the highest relative abundance of genes related to ammonification (*ureC*) on day 10 and 25 (Figs. 2 and S2). Compared to the CK group, on day 10, the AD$_{0.5}$ group exhibits an increase in the relative abundance of certain genes associated with nitrification (*amoC*, *hao*), while others decrease (*nxrA*). On day 25, the relative abundances of both genes, i.e., *hao* and *nxrA*, decrease, while the relative abundance of the gene *amoC* increases (Figs. 2 and S2). Compared to the AOM group, the AD$_{0.5}$ group exhibited higher relative abundances for most of the genes involved in nitrification at two time points, except for the gene *hao* (Figs. 2 and S2). Additionally, the relative abundance of genes

associated with DNRA in the AD$_{0.5}$ group exhibited differential patterns compared to the AOM and CK groups (Figs. 2 and S2). The above results suggest that the relative abundance of N cycling genes is different in the AD$_{0.5}$, AOM, and CK groups, indicating that the presence of DA can alter the relative abundance of sediment N cycling genes.

Although DA can alter the relative abundance of N cycling genes, it is difficult to characterize the effects of DA on N cycling by changes in gene abundance alone. Therefore, KEGG functional module analysis was applied to evaluate the functional pathways associated with N cycling (Fig. S3). The results demonstrate that the pathways of DRNA, nitrification and ammonification are improved in the AD$_{0.5}$ group compared to the AOM and CK groups at two time points, indicating that DA can enhance these three processes. Instead, DA can reduce N$_2$ fixation and denitrification, as they were reduced in the AD$_{0.5}$ group compared with those in the AOM and CK groups. We could not obtain the details of Anammox due to the limitation of the KEGG database. Still, it is reasonable to assume that Anammox was inhibited, because the relative abundances of the most Anammox-related genes are all shown to be descending in the AD$_{0.5}$ group compared to the AOM and CK groups (Figs. 2 and S2). Additionally, it can be seen that the abundances of genes involved in resistance are higher in the AD$_{0.5}$ group than those in the CK and AOM groups on day 10 and 2 (Table S2), which suggests that DA is a stressor for microorganisms. In summary, the above results have further confirmed our speculation that DA can act as a stressor to alter the N cycling in the sediments.

As N cycling is also closely driven by carbon metabolism[26], we next investigated the effect of DA on carbon metabolism via integrative analysis of metagenome and metabolome. The metagenome results reveal that genes with significant changes of carbon metabolism in the three groups are mainly concentrated in the process of carbohydrate metabolism, such as glycolysis, malonate semialdehyde pathway, and tricarboxylic acid (TCA) cycle, etc. (Dataset S1). Moreover, the KEGG functional module analysis shows that glycolysis, TCA cycle and malonate semialdehyde differ in the three treatments. Among them, glycolysis and TCA cycle are inhibited by DA, while malonate semialdehyde is slightly altered by DA (Fig. S4). To confirm the metagenome results, metabolome was utilized to verify the downstream metabolites and metabolic pathways. It can be observed that there are significant differences in metabolites among the three groups on days 10 and 25 (Fig. S5). Through random forest analysis, it is revealed that the metabolites of TCA cycle and glycolysis are also affected by DA,

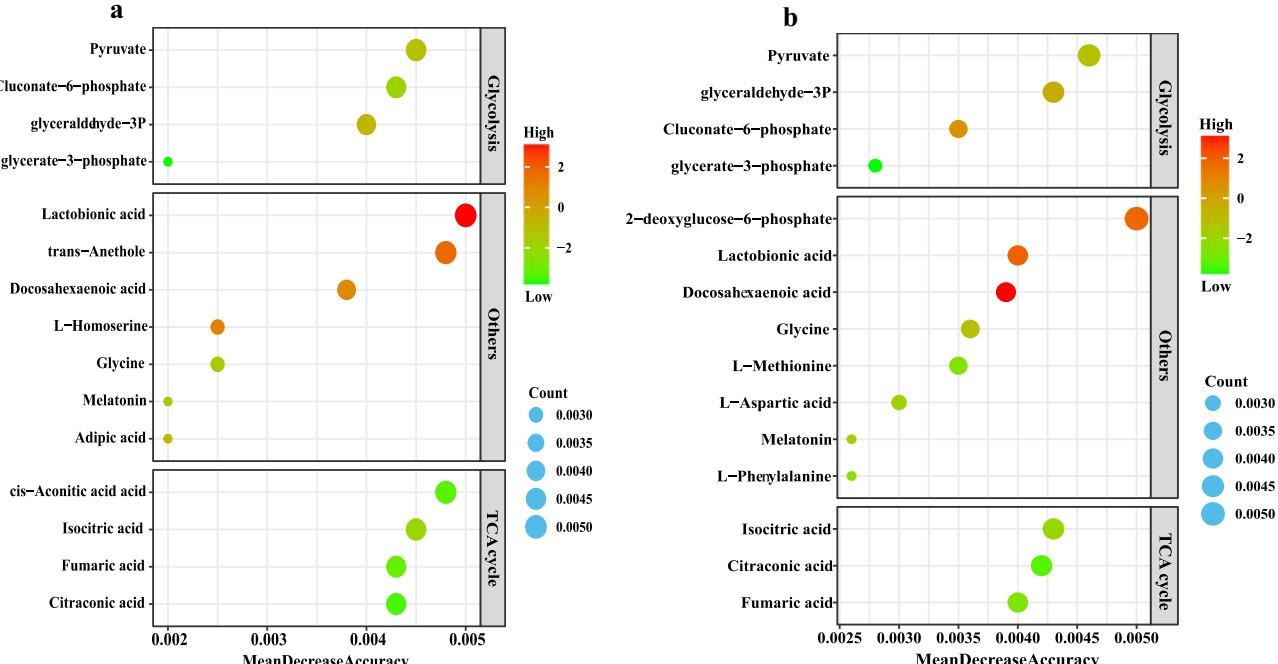

**Fig. 3 | Metabolites with significant differences between $AD_{0.5}$ and CK groups at different times.** Marked metabolites in the $AD_{0.5}$ and CK groups at day 10 (**a**) and 25 (**b**), respectively. The horizontal coordinate of the graph is "Mean Decrease Accuracy", as indicated by the size of the Counts, measuring the importance of metabolites in the "random forest". A higher value indicates a greater importance of the metabolite. The heat map shows the content of the 15 metabolites in the two groups. TCA cycle tricarboxylic acid cycle. $AD_{0.5}$ = modified with algal organic matter and 0.5 mg/l domoic acid, CK = without any modification.

with Fumaric acid, cis-Aconitic acid, pyruvate, etc. identified as the marker differential metabolites (Figs. 3 and S6). Furthermore, the relative abundances of most marker metabolites in the $AD_{0.5}$ group were lower than those in the AOM and CK groups (Figs. 3 and S6), presumably because DA can inhibit the pathways of TCA cycle and glycolysis. Clearly, the integrative analysis of metagenome and metabolome has confirmed that carbon metabolism was altered by DA, with the pathways of TCA cycle and glycolysis being significantly affected. Given that the TCA cycle and glycolysis have been reported as the primary electron supply pathways to drive the N cycling[26,27], their alteration will have cascading effects on N cycling.

**Contribution of DA to the altered N cycling**
Although the presence of DA can alter the relative abundances of functional genes and metabolites associated with N cycling, the contribution of DA and other environmental factors to their alteration still remains unknown. This work utilized Pairwise Spearman's correlation was utilized to uncover the relationships among ecological drivers, N cycling functional genes and metabolites across the sediment microorganisms. It can be seen that DA is strongly relevant to functional genes (Spearman's $r = 0.77$, $p < 0.01$) and metabolites (Spearman's $r = 0.43$, $p < 0.05$). In addition, other factors such as TOC, TON, pH and Ferrous are also closely related to functional genes and metabolites (Fig. 4a). Having illustrated the roles of individual factors in shaping N cycling (genes and metabolites), we next sought to explore the causality and quantify the effects of the drivers via Partial Least Squares Path Modeling (PLS-PM). The final models show satisfactory fit to our data (Fig. 4b), as suggested by the Goodness-of-fit = 0.76. Specifically, DA shows negative impacts on denitrification, Anammox and $N_2$ fixation ($r = -0.98$, $p < 0.01$; $r = -0.51$, $p < 0.05$; $r = -0.73$, $p < 0.01$). Meanwhile, the processes of ammonification, nitrification and DNRA are positively and directly affected by DA ($r = 0.60$, $p < 0.05$; $r = 0.86$, $p < 0.01$; $r = 0.53$, $p < 0.05$). Herein, since AOM is mainly composed of organic C and N[28], its contribution to the altered N cycling is represented by TOC and TON (the changes of TOC and TON content are provided in Fig. S7). AOM shows positive effect on nitrification and Anammox ($r = 0.21$, $p < 0.05$; $r = 0.32$, $p < 0.05$). Unexpectedly, AOM has almost no effect on the denitrification process ($r = 0.07$, $p < 0.05$), which is positively driven by pH ($r = 0.10$, $p < 0.05$). Moreover, other variables are not of great significance on their own, but can obviously improve the model when incorporated together (indicated by the gray dashed line). In summary, our analysis results have not only quantified the impacts of DA on N cycling, but also further verified the above results that DA enhanced nitrification, ammoniation and DNRA, but exerted negative impacts on denitrification, Anammox and $N_2$ fixation.

**Effect of DA on microbial assembly process and N cycling niches**
Besides the above investigations, we further explored the effects of DA on sediment N cycling from the ecological perspective. Generally speaking, the microbial function in environments is generated from the community assembly process[29]. Therefore, the role of DA in shaping the community assembly process was first investigated via neutral community model (NCM) and normalized stochasticity ratio (NST) analysis (Fig. S8). It can be observed that the goodness-of-fit of NCM for the neutral process is low in the groups of $AD_{0.5}$, AOM and CK ($R^2$: 0.261 - 0.309; mobility: 0.04 - 0.054), indicating that community assembly is dominated by deterministic processes (the co-occurrence pattern of microbial communities was non-random). Furthermore, the proportion of deterministic processes in the $AD_{0.5}$ group (0.65) is significantly higher than those in the groups of CK (0.52) and AOM (0.58) (Fig. S8), implying that DA can alter the assembly process of microbial community. Previous studies have shown that microbial assembly process could generate endogenous dynamics among microbes, further driving the variation of ecological functions[30]. In this study, co-occurrence network analysis was conducted to further visualize the linkage between the dynamic succession of the microbial community assembly process and N cycling (KEGG pathways of N cycling) (Fig. 5). In general, the relationships established between the N cycling processes and microbial taxa (at the family level) in the co-occurrence network were different under different treatments. The

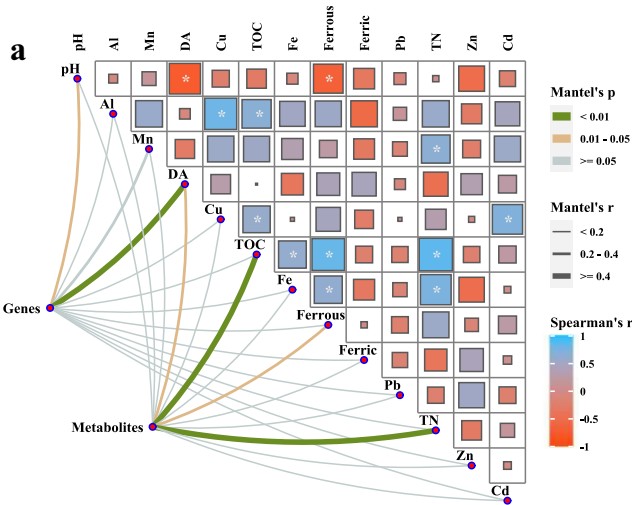

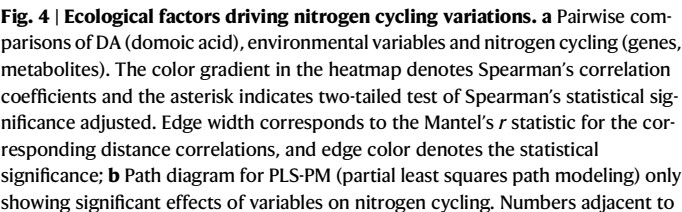

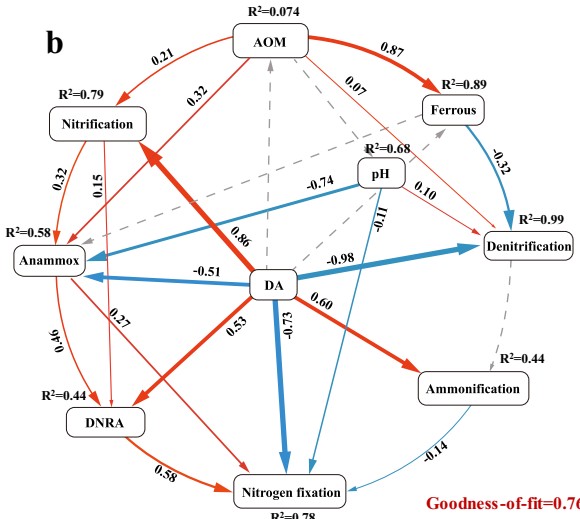

**Fig. 4 | Ecological factors driving nitrogen cycling variations. a** Pairwise comparisons of DA (domoic acid), environmental variables and nitrogen cycling (genes, metabolites). The color gradient in the heatmap denotes Spearman's correlation coefficients and the asterisk indicates two-tailed test of Spearman's statistical significance adjusted. Edge width corresponds to the Mantel's *r* statistic for the corresponding distance correlations, and edge color denotes the statistical significance; **b** Path diagram for PLS-PM (partial least squares path modeling) only showing significant effects of variables on nitrogen cycling. Numbers adjacent to the arrows are standardized path coefficients (*r*), analogous to relative regression weights and indicative of the effect size of the relationship. R2 represents the proportion of variance explained for every dependent variable (in the box) in the model. Blue and red arrows represent significant ($p < 0.05$) negative and positive pathways, respectively. TOC total organic carbon, TON total organic nitrogen, AOM algal organic matter. Nitrogen cycling: ammoniation, nitrification, denitrification, Anammox, N2 fixation, DNRA (dissimilatory nitrate reduction to ammonium).

positive/negative and direct/indirect relationships in the co-occurrence network suggest that the coexistence relationships and cooperative mechanism between microbial taxa and N cycling pathways changed significantly under the intervention of DA. The above results show that the presence of DA can alter the interaction between N cycling and the biotic endogenous dynamics generated by the assembly process, which will feed back different endogenous dynamics to the N cycling, thus causing alterations in the N cycling.

In order to clarify the reason for which the endogenous dynamics generated during the assembly process drive changes in N cycling, the niche distribution of N cycling in different treatments were further investigated, as niche changes are considered to be the key driving factor of alterations in functional traits during the deterministic assembly process[31,32]. Here, we first characterize the niche of the N cycling by identifying the microorganisms with N cycling functions, which is achieved through the co-occurrence network of N cycling genes and microbial taxa (genus level) (Fig. 6, $r > 0.80$, $p < 0.05$). In addition, as the modules of co-occurrence network are generally regarded as niches or functional units that constitute different ecological processes in the community[33,34], co-occurrence networks were modularized to facilitate the visualization of N cycling niches. We highlighted and visualized the major modules with at least ten nodes, and the co-occurrence networks of the AD_{0.5}, AOM and CK groups were divided into 7, 8 and 7 modules, respectively. Through comparing the niches in different phases of the N cycling processes in the three groups, it was found that DA reduced the niches of denitrification, Anammox, and N fixation. For instance, denitrification was distributed in Modules I and II of the AD_{0.5} group. In contrast, denitrification was distributed in Modules II, IV, VI and VII of the AOM group and Modules III, V and VII of the CK group. Besides, DA differentiated niches in nitrification, ammoniation and denitrification processes. For example, DNRA was clustered into Module I, V and VI of the AD_{0.5} group, Module II and V of the AOM group, and Module III and V of the CK group. Taken together, the niches of the N cycling functional microorganisms were significantly changed in the presence of DA, which reflects changes in the functional traits of the N cycling in sediments.

## Discussions

Having proved that DA can regulate the relative abundances of N cycling-related genes to alter the N cycling via macrogenomic analysis (Figs. 2 and S2 and Table S1), we further investigated the mechanism of how DA regulates the expression of N cycling functional genes. In this work, the N cycling processes were found to be significantly associated with the genes of the quorum sensing (QS) system (Fig. S9), which refers to the intercellular communication that allows the adjustments of the gene expression in metabolisms and phenotypes, such as N cycling, biofilm formation and motility[35]. Therefore, DA probably regulates the abundance of N cycling genes through the QS system. Here, the QS genes significantly relevant to the N cycling processes are highlighted ($R > 0.8$, $p < 0.01$). It can be seen that the denitrification and Anammox processes are negatively correlated with genes *las* and *rhl*. In contrast, N_2 fixation, nitrification, DNRA, and ammonification positively correlate with genes *raiR*, *sdiA*, *lux* and *kdpE*, respectively. According to the abundance distribution of the above genes in different groups, genes *las*, *rhl*, *sdiA*, *lux* and *kdpE* are positively regulated by DA, and the gene *raiR* is negatively regulated by DA (Fig. S10). Studies have confirmed that the QS systems of *las* and *rhl*, positively regulated by DA, are able to negatively affect nitrification and Anammox[36,37]. The increase in the abundance of *las* and *rhl* can inhibit denitrification and Anammox, and regulate the expression of the associated genes. The QS systems of *sdiA* and *kdpE*, whose abundances are positively regulated by DA, can positively affect nitrification and ammonification[38,39]. Their elevated abundance will promote nitrification and ammonification, and change the expression of associated genes. The QS system of *raiR*, negatively regulated by DA, also shows to be positively associated with N_2 fixation[40], and a decrease in *raiR* abundance will inhibit N_2 fixation, thus altering the expression of related genes. In this study, the *lux* QS system was shown to be significantly associated with DNRA. Still, no studies on the relationship between *lux* QS system and DNRA have been reported, so this result needs further biological confirmation. In addition, whether DA impacts the cooperation of operon genes, promoter genes, and structural genes in the operon to alter the gene expression of N cycling is required to be further examined.

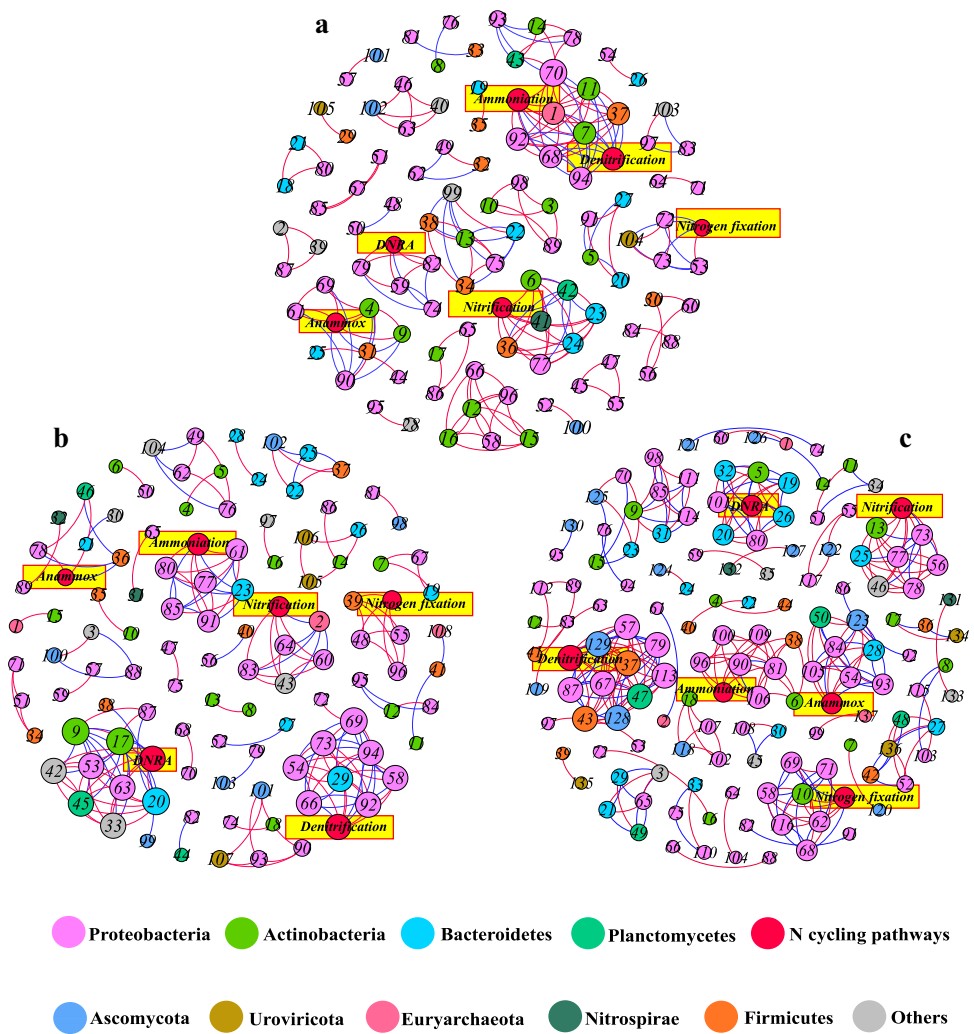

**Fig. 5 | The relationship between microbial community assembly processes and nitrogen cycling.** The linkage between dynamic succession of microbial community assembly process and nitrogen cycling in $AD_{0.5}$ (**a**), AOM (**b**) and CK (**c**) groups. Nodes are colored according to different phylum, and the size of each node is proportional to the number of connections (i.e., degrees). The red link suggests positive covariation between two individual nodes and a blue link reflects negative covariation. Besides, the different numbers represent the microorganisms at different family levels (the information of the nodes is provided in Table S3). Nitrogen cycling: ammoniation, nitrification, denitrification, Anammox, nitrogen fixation, DNRA (dissimilatory nitrate reduction to ammonium). $AD_{0.5}$ = modified with algal organic matter and 0.5 mg/l domoic acid, AOM = modified with algal organic matter only, CK = without any modification.

Having demonstrated that DA can alter N cycling by affecting carbon metabolism (Figs. 3 and S6 and Dataset S1), the potential biological mechanism of how DA regulates the N cycling is further discussed here. It has shown that the electrons for Denitrification, DNRA, N₂ fixation and Anammox are mainly provided by carbon metabolism, and the electrons are mainly donated in the form of nicotinamide adenine dinucleotide (NADH)[26,41], primarily derived from the pathways of glycolysis, pentose phosphate (PP) and TCA cycle[27,42]. Herein, the macrogenomic analysis has confirmed that the pathways of glycolysis and TCA cycle were repressed by DA (Fig. S4), suggesting the inhibition of electron supply for the N cycling. Additionally, following the metabolomic analysis, the relative abundance of pyruvate, fumaric acid and cluconate-6-phosphate, etc. was suppressed by DA (Figs. 3 and S6). Pyruvate is the crucial linker between glycolysis and the TCA cycle in microbial metabolism, so the reduction of pyruvate levels could directly restrain the TCA cycle and then bring downstream effect on the generation of NADH[43]. Besides, studies have shown that metabolites will have a decisive impact on metabolic pathways if they are close to the upstream of the pathway and there are many other metabolites downstream of the pathway[44]. Given that fumaric acid, gluconate-6-phosphate, etc. occupy the upstream in pathways of TCA

cycle and glycolysis (Fig. S11), the decreased levels of these metabolites indicate the suppression of these two pathways, which means, the generation of NADH is reduced in $AD_{0.5}$ group. The diminished electron supply efficiency can adversely impact nitrate reduction, N₂ fixation, and Anammox. However, DRNA, unexpectedly, was still enhanced despite the reduced electron supply efficiency, which may because gene abundance has a greater effect on N cycling than on electron supply. Collectively, the biological mechanisms by which DA alters N cycling can be summarized as follows: (1) DA intervenes in the expression of the genes involved in N cycling by regulating the QS system; and (2) DA regulates the N cycling electron supply efficiency by altering the abundance of local metabolites in the TCA cycle and the glycolysis of C metabolism.

We then attempted to reveal the mechanism of the altered N cycling from an ecological perspective. Studies have demonstrated that microbial taxa will facilitate the adaptive reconstruction of co-occurrence patterns and introduce new interactions with microorganisms in the face of stressors. These newly generated endogenous dynamics will exert downstream effects on community functions[30]. Such an ecological succession mechanism was also observed in our work. Under the intervention of DA, the community assembly

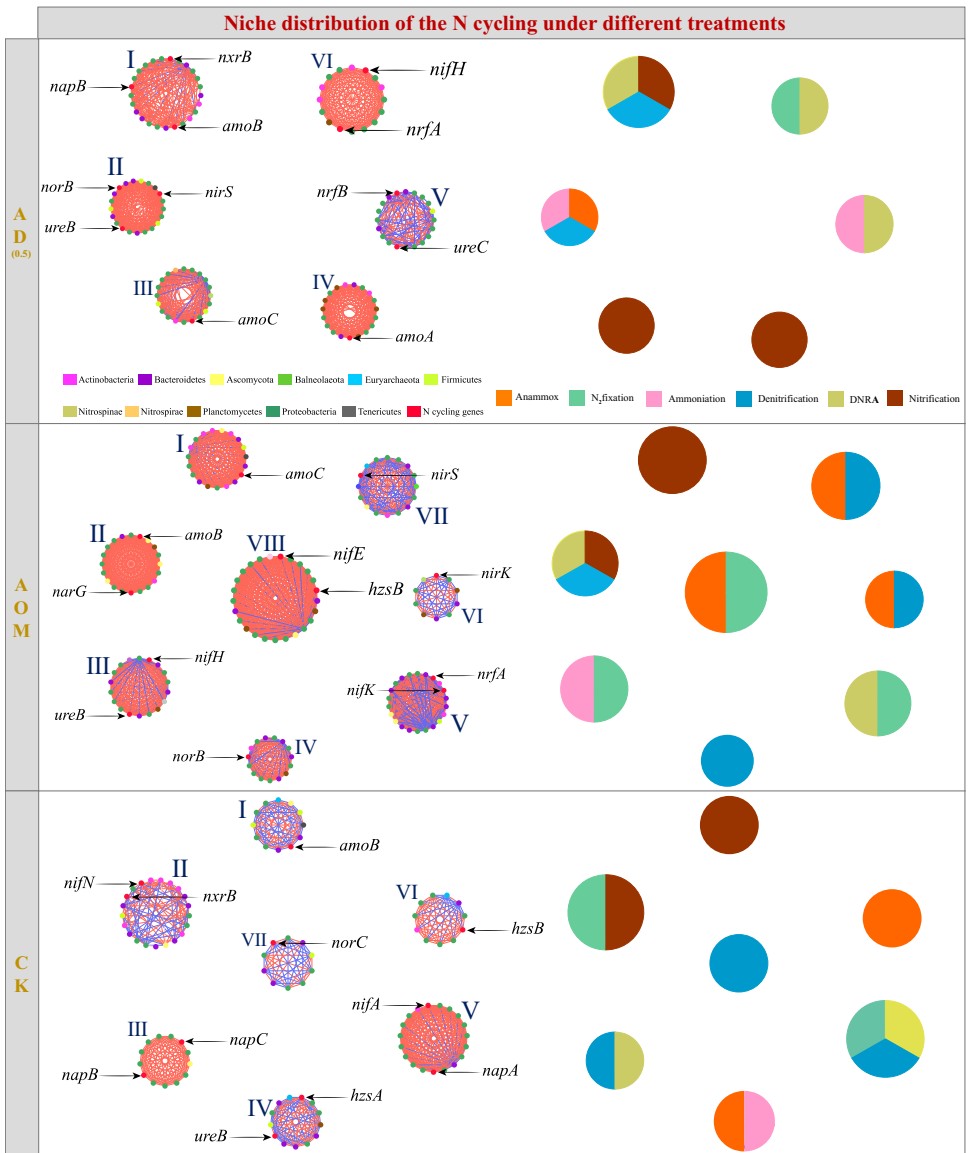

**Fig. 6 | Modules and niches of AD_0.5, AOM and CK group.** Colors of nodes indicate different major phylum and nitrogen cycling genes; pie charts represent the composition of N (nitrogen) cycling niches. Inside each module, a red link suggests positive covariation between two individual nodes, whereas a blue link reflects negative covariation. Nitrogen cycling: ammoniation, nitrification, denitrification, Anammox, nitrogen fixation, DNRA (dissimilatory nitrate reduction to ammonium). AD_0.5 = modified with algal organic matter and 0.5 mg/l domoic acid, AOM = modified with algal organic matter only, CK = without any modification.

mechanism is altered, leading to significant changes in the coexistence relationships and cooperative mechanisms between microbial taxa and N cycling pathways (Figs. 5 and S8). Exampling the denitrification in the AD_0.5 group (Fig. 5), it is driven by multiple dynamics in the co-occurrence network, directly or indirectly. For example, Sulfuricellaceae[23] (negative) and Pseudomonadaceae (positive)[45] can act directly on denitrification, supplying positive and negative drivers to the denitrification, respectively. In addition, these microorganisms directly associated with the denitrification are also regulated by other taxa in the co-occurrence network, exerting an indirect driving effect on the denitrification. Likewise, the other N cycling processes in the AD_0.5 group are also regulated this way. The different symbiotic patterns in the three treatments can alter the cooperation mechanism between microbial taxa and N cycling, thus leading to further changes in the process of N cycling.

The changes in the biotic endogenous dynamics of the co-occurrence network further resulted in a shift in the niches of the N cycling functional microorganisms, which was shown to be responsible for the altered sediment N cycling (Fig. 6). Under the action of DA, an

environmental filter factor, the generalists in the network may transform their survival mode and then migrate from their original niche to another. In the present work, the microorganisms involved in denitrification and DNRA (two competing nitrate reduction metabolic pathways) were shown to switch their metabolic mode between these two pathways under different conditions[46]. This may explain why DA alters these two functions, consistent with the previous ecological finding that the environment selects the functions rather than microbes[47]. Moreover, some microorganisms, resistant to external pressure, can use the available resources in the environment to generate more niches. For instance, in the AD_0.5 group, the increasing concentrations of organic N and NH_4^+ and the correspondingly increased niches in ammoniation and nitrification suggest intensified ammoniation and nitrification. Presumably, the microorganisms involved in these two processes may be insensitive to DA, and the surplus substrate organic N and NH_4^+ (the electron donor for nitrification) will offer more niches for them. In general, the specialists in the network always have low environmental tolerance and particularly high sensitivity to environmental change and variability[48], and are

difficult to be replaced. It is speculated that the N fixer and Anammox, which both belong to typical specialists in the ecosystem[49], are vulnerable to DA stress, and the niches are thus squeezed by the external pressure, resulting in their functional traits being weakened in the system. To sum up, we explored the mechanism by which DA alters the N cycling from the ecological perspective. As a stressor, DA changes the microbial assembly process, leading to different interactions between endogenous dynamics and N cycling, further altering the niche of N cycling function microorganisms and ultimately causing shifts in N cycling. It is noteworthy that DA can be converted into other compounds (Fig. S12); therefore, changes in the N cycling should be attributed to DA and its degradation products. This indicates that the composite effects of DA altered the N cycling in sediment.

As a widely-distributed red tide algal toxin with intense neurotoxicity, DA has attracted worldwide attention[5,7,15]. However, the existing studies are just confined to its toxicological effects on marine animals, without any attempts to reveal the effects of DA on marine microbes, which is the engine of biogeochemical cycling[13,14]. In this work, we have uncovered that DA can act as a stressor to alter marine N cycling, thus bridging the knowledge gap of the ecological impact of DA on the marine ecosystem. Besides the functionality, whether the ecosystem will be in a new steady state after the algal bloom should also be investigated. If not, will the sediments recover to its original state over time? This obviously requires to be addressed in future field studies. Finally, we recommend implementing necessary measures to prevent the outbreak of HABs. For example, enhancing control and monitoring of non-point sources of phosphorus and nitrogen in the offshore environment[50]. Additionally, establishing a developed HAB prediction method is a crucial strategy for identifying the formation of algal blooms and providing early warnings to local authorities for preparation[51]. Governments could also pursue more proactive initiatives, such as conducting public education activities, to avoid, minimize, and address the occurrence of HABs, while effectively communicating with the public about HABs.

## Methods

### Sample collection and materials preparation

The sediments and seawater were collected from the top 2 cm of the Heishijiao coast, located along the Yellow Sea, Dalian, China at low tide in April 2022. The sampled sediments were thoroughly homogenized, with the interstitial water and large debris simultaneously removed. Then, the homogenized sediments were transferred to the laboratory within 30 min for microcosm establishment. It was found that there was no DA in the sampled sediments (the DA detection method is provided in Text S1). AOM was made from the cell lysates of *P.americana* (a non-toxic algae belonging to the genus *Pseudo-nitzschia*) and *Skeletonema costatum*, based on the method referenced from previous studies[52]. Here, *skeletonema costatum* was used to replace other diatoms, as it is the dominant species during HABs in addition to *Pseudo-nitzschia*. These two types of algal cells were purchased from the Institute of Hydrobiology (Chinese Academy of Sciences, Wuhan). The number of *P. americana* was set at $2.0 \times 10^7$ cells $g^{-1}$ sed according to previous field investigations[9]. Since the number of *Pseudo-nitzschia* was reported to be 50% of the total algal cells[53], the number of *Skeletonema costatum* was also set as $2.0 \times 10^7$ cells $g^{-1}$ sed. In addition, DA standard was purchased from Toronto Research Chemicals (purity ≥ 95%).

### Setup of microcosm

The microcosms were established according to the guidelines of OECD 308[54]. Specifically, ~300 ml of seawater and 200 g of wet sediment were added to sterilized 500 ml flasks, allowing to establish an oxic/anoxic gradient down the microcosms prior to experiments. After that, the microcosms were covered with aluminum foil to prevent evaporation and then maintained on a rotary shaker ($20 \pm 1\,°C$) with a

speed of 100 r $min^{-1}$ in darkness. After a week of acclimation under experimental conditions, the experiment for exploring the effect of DA on sediment N cycling was carried out. The experimental setup is as follows: (1) Microcosms with modified AOM and DA were set as the experimental group. In accordance with a previous field work, which monitored the distribution of DA in marine benthic environment for 15 consecutive years[9], the DA released by the decay of *Pseudo-nitzschia* cells is primarily present in the aqueous phase, i.e., in the form of dDA; the concentration of dDA is commonly about 0.6 mg $l^{-1}$, and its maximum concentration can reach 4.2 mg $l^{-1}$[9]. Therefore, in the present study, the concentrations of dDA in microcosms were set as 0.1, 0.5 and 1 mg/l, which were named as $AD_{0.1}$, $AD_{0.5}$ and $AD_{1.0}$, representing low, medium and high concentrations, respectively. It should be mentioned that the added dDA can also be adsorbed by the sediment. The maximum concentrations of DA in the sediments of the three groups were 11.51, 15.75 and 21.11 ng/g wet sed, respectively (the same order of magnitude as the previous field detection concentrations (levels of ng/g wet sed)[10]). Most of the DA in the system was still present in the form of dDA (92.3%, 97.9%, 98.6%), with the concentrations of 0.0923, 0.4895 and 0.9860 mg/l, respectively. Overall, we believe that the exposure concentrations we set are environmentally relevant, particularly for the foundational work. (2) The microcosms without any modification served as the CK group. (3) The microcosms with only AOM (the AOM group) and 0.5 mg $l^{-1}$ DA (the DA group) modified were mainly to provide support for statistical analysis. Nine biological replicates were set for each treatment. Three microcosms were randomly selected for destructive sampling on day 10 and day 25 of the experiment in AOM, $AD_{0.5}$ and CK groups. Specifically, triplicate cores were randomly collected at three locations within each microcosm, and then the three cores were homogenized and centrifuged, and the supernatant water was removed using a pipet for the assays of potential N transformation rates (Text S2), heavy metal content, total organic carbon (TOC) and total organic nitrogen (TON) in sediments (Text S3). Additionally, the AOM, $AD_{0.5}$, CK and DA groups were used for the sequencing of metagenome and metabolome on day 10 and day 25. The remaining samples from the AOM, $AD_{0.5}$ and CK groups were applied for the detection (every 3 days) of dissolved inorganic nitrogen (DIN, $NO_3^-$, $NO_2^-$, $NH_4^+$) in the overlying water.

### DNA extraction and metagenomic sequencing

DNA was extracted from sediments samples using CTAB method. DNA degradation degree, potential contamination and DNA concentration was measured using Agilent 5400. Then, sequencing library was generated using NEBNext UltraTM DNA Library Prep Kit for Illumina (NEB, USA, Catalog#: E7370L) following manufacturer's recommendations and index codes were added to each sample. Briefly, genomic DNA sample was fragmented by sonication to a size of 350 bp. Then DNA fragments were endpolished, A-tailed, and ligated with the full-length adapter for Illumina sequencing, followed by further PCR amplification. After PCR products were purified by AMPure XP system (Beverly, USA). Subsequently, library quality was assessed on the Agilent 5400 system (Agilent, USA) and quantified by OPCR (1.5 nM).

### Sequencing and bioinformatics analysis

The qualified libraries were pooled and sequenced on Illumina platforms with PE150 strategy, according to effective library concentration and data amount required. Then, MEGAHIT software (v1.2.9) was used to assemble clean reads after de-host gene and obtain contigs. The de-redundant protein sequence (corresponding to the de-redundant gene nucleic acid sequence) was blast to the EggNOG database (v5.0, http://eggnog5.embl.de/#/app/home) to obtain the KEGG (v102.0, https://www.kegg.jp/kegg/), GO (released in 2023-05-16 https://geneontology.org/) and COG (v2020, https://www.ncbi.nlm.nih.gov/COG/) annotation information of the protein. Then, the de-redundant protein

sequences are blast to the CAZy (v2020-05, http://www.cazy.org/) database to obtain the annotation information of CAZy by DIAMOND software (v0.8.22). Finally, according to the abundance table of the de-redundant gene and the annotation information of each database, the de-redundant gene abundance (TPM) annotated to the same gene family in the database is summed for each database, and the failed de-redundant genes are screened out to obtain the relative abundance table of each database gene family. The KEGG Module (v102.0) database serves to classify genes with analogous functionalities into KEGG Orthology (KO) groups, thus assigning each KO to a distinct function. In biological processes, functional pathways often require the coordinated action of multiple functional units. In such cases, the KEGG Module database integrates multiple KOs, enabling the identification of functional activity at the pathway level. Besides, due to the limitations of the KEGG database, we additionally annotated Anammox-related genes via the NCyc (https://github.com/qichao1984/NCyc) database using Diamond (v2.0.14)[55]. It is important to note that gene data employed for calculating changes in gene relative abundance and assessing gene abundance correlations are associated with the rarified and logarithmic transformation respectively.

### Taxonomy annotation
Kraken2 (v2.0.7-beta) and the self-build microbial database (Sequences belonging to bacteria, fungi, archaea and viruses were screened from NT nucleic acid database and RefSeq whole genome database of NCBI (https://ftp.ncbi.nih.gov/blast/db) were used to identify the species contained in the samples. Then, Bracken (v2.0) was used to predict the actual relative abundance of species in the samples.

### Untargeted metabolomics
The samples (1 ml) were freeze-dried and resuspended with prechilled 80% methanol by well vortex. Then the samples were incubated on ice for 5 min and centrifuged at $15,000 \times g$, 4 °C for 15 min. Some of supernatant was diluted to final concentration containing 53% methanol by LC-MS grade water. The samples were subsequently transferred to a fresh Eppendorf tube and then were centrifuged at $15,000 \times g$, 4 °C for 15 min. Finally, the supernatant was injected into the LC-MS/MS system analysis.

### Metabolite identification
The raw data files generated by UHPLC-MS/MS were processed using the Compound Discoverer 3.1 (CD3.1, ThermoFisher) to perform peak alignment, peak picking, and quantitation for each metabolite. The main parameters were set as follows: retention time tolerance, 0.2 min; actual mass tolerance, 5 ppm; signal intensity tolerance, 30%; signal/noise ratio, 3; and minimum intensity, et al. After that, peak intensities were normalized to the total spectral intensity. The normalized data was used to predict the molecular formula based on additive ions, molecular ion peaks and fragment ions. And then peaks were matched with the mzCloud (https://www.mzcloud.org/), mzVault and MassList database to obtain the accurate qualitative and relative quantitative results.

### Statistical analyses
Statistical analyses were performed using R (v4.2.1). Tukey's test is used for significance analysis of percentage differences between treatments. Correlations of N cycling genes and metabolites with environmental factors were analyzed by Mantel test. Environmental factors correlated with N cycling genes and metabolites were selected for the partial least squares path modeling (PLS-PM). A PLS-PM was used to analyze the effects of DA, AOM, and environment factors on N cycling (genes, metabolites). The PLS-PM model was constructed using "plspm" package (v0.5.0) with the Goodness-of-fit (GOF) index to judge the model's overall fitting degree. The Neutral Community Model (NCM) is used to identify the process of community assembly process (package "stats4" (v4.3.0), "minpack.lm" (v1.2-3) and "Hmisc" (v5.1-0)). To quantify stochasticity in the microbial assembly process, the normalized stochasticity ratio (NST) was calculated using the "NST" package (v3.1.10) with Bray-Curtis dissimilarity, 999 randomization and "PF" null model. The boundary point was 50%, where NST < 50% indicates that the deterministic process is the dominant process and NST > 50% indicates that the stochastic process is the dominant process[56]. Co-occurrence network analysis was conducted using the "Hsmic" package (v5.1-0). Microbial taxa at the family and genus levels, with relative abundances exceeding 0.05%, were employed in constructing networks alongside N cycling pathways and N cycling genes. The absolute value of Spearman's correlation coefficient (|ρ|) > 0.8 with $p$ value < 0.05, which stands for the strong pairwise correlations among microbial taxa, N cycling genes and pathways, was selected for the network. The network properties were calculated using "igraph" package (v1.4.3), and the networks were visualized using Gephi (v0.9.6) and Cytoscape (v3.9.1). Furthermore, the co-occurrence network was modularized and visualized using the MCODE plugin within Cytoscape. The random forest model is used to select the marker metabolites, completed through the bioinformatics analysis website (bioinformatics.com.cn). Correlations between QS genes and N cycling pathways were calculated based on the "correlation plot" of Originpro 2023.

### Reporting summary
Further information on research design is available in the Nature Portfolio Reporting Summary linked to this article.

## Data availability
The macrogenomic data generated in this study have been deposited in National Genomics Data Center under accession code PRJNA912632. Moreover, the databases used to obtain the annotation information are as follows: EggNOG database to obtain the KEGG, GO, COG, and CAZy database. NT nucleic acid database and RefSeq whole genome database of NCBI can be found at https://ftp.ncbi.nih.gov/blast/db. NCyc database is available at https://github.com/qichao1984/NCyc. The authors declare that the remainder of data that support the findings of this study are available within the article and source data file (https://doi.org/10.6084/m9.figshare.22817291)[57]. Source data are provided with this paper.

## Code availability
The in-house R scripts and relevant data used to generate figures of this study are provided with this paper and publicly available on GitHub (https://doi.org/10.5281/zenodo.8431607)[58].

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

## Acknowledgements

We acknowledge the financial support of the National Natural Science Foundation of China (No. 42177375 to J.W.) and (No. 21876018 to J.W.).

## Author contributions

Z.L. (wrote the entire paper) and H.Y.(wrote the method section) wrote this manuscript, Z.L. and H.Y. conducted the experiments and data analyses. J.W. planned research. J.W. and J.F. involved in supervised all analyses, data interpretation and discussion as the Project Leader. M.D. and Y.J. assisted in conducting some experiments and sample analyses.

## Competing interests

The authors declare no competing interests.
