## [Peer Review File NEW · Nature Communications]

Marine Toxin Domoic Acid Alters Nitrogen Cycling in SedimentsReviewer #1 (Remarks to the Author):

"Marine Toxin Domoic Acid Alters Nitrogen Cycling in Sediments" written by Li and co-authors test their hypothesis that DA is capable of affecting nitrogen cycling in sediments. This work is highly intriguing. Current research hints to the possibility the DA have some effect on microorganisms, however, it is still unknown whether this is true and if so, in what way. Additionally, DA sediment affects in highly important to understanding given the high accumulation and toxin nature.

I have two main questions about the study. The first is dissolved vs. particulate DA. Throughout the manuscript it appears that particulate form is only being considered. There is a high probability that dissolved DA is present within the pore waters in sediments but this is never discussed. The second is the idea that DA presence is altering N cycling. The authors make it seem that there is a fluctuation of DA within the sediments that would then affect the N cycle. However, I would assume that a consistent amount of DA is accumulated within the sediments, unless there is a degradation process, I have no considered. As such, would it be more appropriate to say that changes in DA concentration influence the variation in N cycling? Or am I missing something?

More Specific Comments:

Line 37: "... that <is> produced by ..."

Introduction:

I think it is important to distinguish between particulate and dissolved forms of DA. My interruption of the authors is that sediment DA is primarily found from particulate aggregation in marine snow. Is there any evidence that dissolved forms of DA are present within pore waters? We know that dissolved DA can also affect marine organisms like filter feeders, is this considered as well?

Methods:

It was not clear which species of *Pseudo-nitzschia* was used within the experiments. Much research has shown that different species can have specific effects. Please clarify.

I am confused as to how the authors separate particulate from dissolved forms. I assumed that all DA within the project is particulate, is that correct? If not, then something is needed for dissolved forms or at least a comment about it within the discussion.

Results:

Figures: All figures are difficult to read. In particular, the axis labels are very hard to read, please increase their size.

Figure 2: I have found that reading red text on a background color can be quite difficult. I suggest changing the color to black- making it easier to read.

I am a little confused why a random forest was used to determine which marker metabolites were most common. Please justify the use of the random forest. I feel like other more specific statistics could be used.

Discussion:

There is a lot of logic (A and B lead to C and then lead to D) happening with the discussion. This is good to try and follow the authors thought process. However, there is quite a lot of that, in my opinion. I would appreciate a few more citations to help simplify the logic. Also, I think a caveat paragraph would be great to see. Maybe this would be a great place to include my main questions within the manuscript.

Reviewer #2 (Remarks to the Author):

The manuscript attempts to quantify the effect of Domoic Acid (DA) on nitrogen cycle processes in sediments. Earlier work by others showed that diatoms produce a lot of DA, and that it can be transported to sediments by sinking organic material from diatoms growing in surface waters.

Thus this project addressed a logical question, so the DA that makes it to ocean sediments affect the biogeochemical processes in those sediments.

The experiment was a straightforward incubation design with adequate replication and state of the art analysis of everything from standard water chemistry to metabolomics.

The text needs a good English editor. I was not able to understand some of the text, so I cannot not make a detailed review at this time. In hopes that the presentation and content can both be improved at the next stage, I will provide as much of a review as is possible at a fairly high level, although some of my concerns might be mitigated by a better understanding of the presentation.

I have two serious scientific concerns.

1. The level of Domoic Acid (DA) additions to the microcosms ranged from 0.1mg/L to 1mg/L. A review of the literature, including the two papers cited in this manuscript as the basis for the choice of the DA addition concentration, finds that maximum reported seawater concentrations ranged from 0.220 µg/L (0.00022 mg/L) in the central Pacific (Silver et al. 2010) to ~6 µg/L (0.006mg/L) in Santa Barbara Basin (Sekula-Wood et al. 2009). The DA additions to the DA0.5 microcosm exceeds these environmental levels by at least 83-fold.

On the other hand, DA concentration has been measured in sediment traps at up to 163 µg/g dry wt of sediment (Sekula-Wood et al. 2009). If the final concentration of DA in the DA0.5 microcosm was 250 µg in 500 ml (0.5 mg/L in 500 ml, including 200g of sediment), that's 1.25 µg DA/g sediment. By that comparison, the DA additions in this experiment are far below what might be expected to occur in sediments underlying a diatom bloom.

Could the authors please explain their effective DA concentrations? Was the concentration actually measured in the experimental microcosms? The method is cited, but no measurement data are presented. Does the final concentration (e.g., 0.5 mg/L) refer to the 300 ml of water or the total volume of the microcosm including sediments? If my calculations are correct, it seems unlikely that organisms would ever be subjected to such high bulk water concentrations, but they could experience a large flux of DA to the sediment when bloom debris is deposited. I'm having a hard time figuring out if the experimental additions were high or low. The authors refer to them as trace level, but it's not clear if that is the case.

2. The "abundances" of all the genes are often referred to in the text as "abundances", not "relative abundances". This is very misleading. This kind of data is by nature compositional and it's really hard to translate it to absolute abundances. However, the authors appear to be treating the relative data as absolute abundances, when they refer to changes in abundance of the various N cycle genes (most of page 10, e.g.), etc. The supplemental text does not provide much information on how the abundances were calculated. Were they normalized to total read number only? Or rarified to account for the total number of reads in individual samples? Or better yet, use the centered log transformation (Aitchison 1986). Table S1 provides the relative abundance numbers from which the treatment effects shown in Figure 2 were calculated. But the actual numbers in the Table need to be explained. If it's simple relative abundance as a fraction of the total reads, these would all be less than 1, or if expressed as a percentage, less than 100. The range of values in Tables S1 and S2 is from zero to >300. We need to know what these numbers are. (I was able to use this table to get the numbers that appear in Figure 2, it's just not clear where the data in Table S1 came from.)

I assume the gene relative abundance data were also the basis of KEGG pathway analyses, so this "abundance" assumption is also important to acknowledge and deal with here. Correlation analyses are also problematic with proportional data – the Mantel test probably used a Pearson correlation matrix, which is not appropriate.

This concern affects all the network analyses as well. It's possible that a more rigorous statistical analysis of the compositional relative abundance data would not change the interpretation very much, but the proportional nature of the data must be acknowledged and the implications addressed. I.e., it's a zero sum game. You cannot tell that one gene increased in absolute abundance because all abundance calculations depend on everything else in the sample. Therefore

standard correlations don't make sense.

Other comments:

Planetary boundary threat? I will need some convincing.

Figure 3. We need more information here. What are the units for "counts"? Were the metabolomic data analyzed in triplicate, as was done for the metagenomic analysis? The methods imply that triplicates were analyzed. If so, can we have some indication of statistical significance of the count data?

Table S1. Please state in the caption that these data refer to e.g., DA0.5 day 10. "Treat_10" did not immediately make sense to me.

Figure S2. It would be good to provide units for the rates somewhere on this page. The units are given in the methods, but should be noted here. It seems to me that these data are central to the interpretation of the main point, that actual rates of important N cycle processes were affected by DA. However, these are potential rates (according to the methods, in which very high substrate levels were added to very small incubations). So the significance of DA effects on potential vs in situ rates should be addressed in the text.

These plots are not quantitative enough. I suggest a column plot showing each rate and the error for the three replicate incubations for each experiment. And these data should be in the main text, perhaps as additional plots in Figure 1. The plots should contain information on statistical comparisons between rates among the different treatments and days. The colored dots just don't contain enough information to allow the reader to evaluate the treatment effects.

L 233 Summarily – please explain the significance of these particular metabolites. This statement is too general to mean anything.

L374

The metabolite data are interpreted here to show repression of specific pathways – these are central pathways in the metabolism of just about every cell. Does this mean that the DA concentration was so high that all basic metabolism was affected? The phrase "electron provision" is not one with which I am familiar. Does this mean electron transport? Respiration based on redox transfers? I can't make sense of this argument.

Figure 4 is very nice and contains a lot of information. The caution about applying simple Spearman correlation analyses to the gene abundance data applies, however. What do the R2 values refer to in Figure 4b? Each R2 is adjacent to a box, but each box has a different number of arrows connecting to the box. If the R2 applies to one of the arrows, that should be stated and made clear in the figure. E.g., for the boxes at the top and bottom of the figure, the R2 numbers appear to apply to the box, not to a particular arrow. Even if the R2 did apply to an arrow, what does the R2 of a path coefficient (r) mean?

Figures 5 and 6 are perhaps attractive but impossible to read. One cannot see the taxon id in the left panel of Figure 6. In figure 5, the names of the nodes are unreadable with the low resolution files (worse in the supplemental network figures). Might be okay reading a high resolution pdf file online.

Dear editor,

We are pleased to submit the revised manuscript (ID: NCOMMS-22-53079). We are very much thankful for your fruitful comments, which will further improve our manuscript. As indicated below, we have checked all the specific and general comments provided by reviewers and made changes in response to those comments in highlight in the revised manuscript. The explanation of the revision is as follows:

Response to the Comments of Reviewer #1:

General comments:

“Marine Toxin Domoic Acid Alters Nitrogen Cycling in Sediments” written by Li and co-authors test their hypothesis that DA is capable of affecting nitrogen cycling in sediments. This work is highly intriguing. Current research hints to the possibility the DA have some effect on microorganisms, however, it is still unknown whether this is true and if so, in what way. Additionally, DA sediment affects in highly important to understanding given the high accumulation and toxin nature.

Response: Thank you very much for your comments. According to your suggestion, we have taken great care to respond to each comment and make necessary changes throughout the manuscript as needed. We are grateful for these critical comments, which have improved the presentation of our research findings. Again, thank you very much for your suggestion.

Specific comments:

-I have two main questions about the study. The first is dissolved vs. particulate DA. Throughout the manuscript it appears that particulate form is only being considered.

There is a high probability that dissolved DA is present within the pore waters in sediments but this is never discussed.

Response:

We appreciate the question and apologize for not clearly explaining the different forms of DA concentrations in the system. First of all, we should clarify to you the distribution of DA in the marine benthic environment. In the wake of phytoplankton bloom, the DA-contained algal cells can migrate with the marine snow to the seafloor, and these algal cells usually reach the deep seafloor relatively intact^{1,2}. After the lysis of these algal cells, DA will be released from the algae cells. Due to the high solubility of DA in seawater³, most of the DA will be dissolved in the seawater and exist in the dissolved state of DA (dDA); only a small fraction of dDA will be adsorbed by the sediment (<10%)⁴. Therefore, in this work, the DA concentration was set based on the concentration of dDA in the environment.

Secondly, the selection of dDA concentrations was based on the data from 15-year continuous fieldwork. This work has confirmed that the concentration of dDA is commonly about 0.6 mg/L, and the maximum concentration of dDA could reach 4.2 mg/L⁴. Therefore, the concentrations of dDA in microcosms are set to 0.1, 0.5, and 1 mg/L, representing low, medium, and high concentrations, respectively. Moreover, this fieldwork also demonstrated that DA concentration in sediments was several orders of magnitude less than those detected in sinking marine snow, further indicating that when DA is released from algae cells, DA is mainly present in the form of dDA.

Finally, since another focus of this work is to investigate the biotransformation of DA

in the system (the DA biodegradation time and biodegradation products are provided in Lines 472-474 and Fig. S10, respectively), we monitored the DA concentrations in seawater and sediments throughout the experiment. The added dDA (0.1, 0.5, and 1 mg/L) could be adsorbed by the sediment, and the maximum concentrations of DA in the sediments in the three groups were 11.51 ng/g wet sed, 15.75 ng/g wet sed, and 21.11 ng/g wet sed, respectively (the same order of magnitude as the previous field detection concentrations (levels of ng/g wet sed) ⁵). Most of the DA in the system was still present in the form of dDA (92.3%, 97.9%, 98.6%), with concentrations of 0.0923, 0.4895, and 0.986 mg/L, respectively. Overall, the exposure concentrations of DA in the seawater and sediment were both environmentally relevant. We have redescribed the settings of DA concentration and the DA concentrations in seawater and sediment in the Introduction and Methods (Lines 49-56; Lines 443-458). Thank you again for your comments!

- The second is the idea that DA presence is altering N cycling. The authors make it seem that there is a fluctuation of DA within the sediments that would then affect the N cycle. However, I would assume that a consistent amount of DA is accumulated within the sediments, unless there is a degradation process, I have not considered. As such, would it be more appropriate to say that changes in DA concentration influence the variation in N cycling? Or am I missing something?

Response:

Thank you for your comment. In fact, we have been monitoring the changes of DA concentration in seawater and sediments, and found that DA can be biotransformed. The degradation of DA started on day 8 and was completely degraded on day 36, and

three major DA degradation products were detected in the system (Fig. 1). Therefore, the alteration of sediment N cycling is attributed to the effect of DA and its degradation products. Thank you again for your comments. The result of DA biotransformation was presented in Lines 472-474.

Fig 1. Biotransformation products of DA

-Line 37: "... that produced by ..."

Response:

Thank you for your comments, this has been modified accordingly. In addition, after the first review, this manuscript has sought the help of professional English editors in order to be reviewed by the referees appropriately.

-Introduction: I think it is important to distinguish between particulate and dissolved forms of DA. My interruption of the authors is that sediment DA is primarily found from particulate aggregation in marine snow. Is there any evidence that dissolved forms of DA are present within pore waters? We know that dissolved DA can also affect marine organisms like filter feeders, is this considered as well?

Response:

Thank you for pointing this out! We have modified the Introduction based on your comments (Lines 49-56). According to previous fieldwork, when DA released from algal cells, it is mainly enriched in the aqueous phase, i.e., in the form of dDA⁴. The average concentration of dDA is commonly about 0.6 mg/L, and the maximum concentration can reach 4.2 mg/L. In addition, a small fraction of DA that released from the algal cell would be adsorbed by the sediment (<10%), and its concentration could reach the level of ng/g wet sed⁵. Therefore, the concentrations of dDA in microcosms are set to 0.1, 0.5, and 1 mg/L, representing low, medium, and high concentrations, respectively. Besides, the added dDA could also be adsorbed by the sediment. The maximum concentrations of DA in the sediments among the three groups were 11.51 ng/g wet sed, 15.75 ng/g wet sed, and 21.11 ng/g wet sed, respectively (the same order of magnitude as the previous field detection

concentrations ⁵). Most of the DA in the system was still present in the form of dDA (92.3%, 97.9%, 98.6%), with concentrations of 0.0923, 0.4895, and 0.986 mg/L, respectively. Based on the above data, the exposure concentrations of DA were environmentally relevant in this work.

-Methods: It was not clear which species of *Pseudo-nitzschia* was used within the experiments. Much research has shown that different species can have specific effects.

Please clarify.

Response:

Thank you for your comment. The *Pseudo-nitzschia* species used in the experiments was *P.americana*, a non-toxic algae. The reasons are as follows: the use of non-toxic algae allows us to obtain accurate DA concentrations in the system, and we simultaneously set up groups of AOM (without DA addition) and CK as control groups, which will help us to eliminate the interference of AOM when analyzing the effects of DA on sediment N cycling. We have added the names of the *Pseudo-nitzschia* species to the methods (Lines 425-426).

-I am confused as to how the authors separate particulate from dissolved forms. I assumed that all DA within the project is particulate, is that correct? If not, then something is needed for dissolved forms or at least a comment about it within the discussion.

Response:

Thank you very much for your comments. We have clarified the concentration of DA in different states of the system in the Method (Lines 443-458). According to previous fieldwork, when DA released from algal cells, it is mainly enriched in the aqueous

phase, i.e., in the form of dDA ⁴. The average concentration of dDA is commonly about 0.6 mg/L, and the maximum concentration can reach 4.2 mg/L. In addition, a small fraction of DA that released from the algal cell would be adsorbed by the sediment (<10%), and its concentration could reach the level of ng/g wet sed ⁵. Therefore, the concentrations of dDA in microcosms are set to 0.1, 0.5, and 1 mg/L, representing low, medium, and high concentrations, respectively. Besides, the added dDA could also be adsorbed by the sediment. The maximum concentrations of DA in the sediments among the three groups were 11.51 ng/g wet sed, 15.75 ng/g wet sed, and 21.11 ng/g wet sed, respectively (the same order of magnitude as the previous field detection concentrations ⁵). Most of the DA in the system was still present in the form of dDA (92.3%, 97.9%, 98.6%), with concentrations of 0.0923, 0.4895, and 0.986 mg/L, respectively.

-Results: Figures: All figures are difficult to read. In particular, the axis labels are very hard to read, please increase their size.

Response:

Thank you very much for your comments, all figures been modified accordingly (Fig 1-6). We have made every effort to make the pictures easy to read. If you still cannot see clearly, please enlarge the image. All figures are vector images, enlarging the figures will not affect their resolution. Thank you again for your valuable suggestion!

- Figure 2: I have found that reading red text on a background color can be quite difficult. I suggest changing the color to black- making it easier to read. I am a little confused why a random forest was used to determine which marker metabolites were most common. Please justify the use of the random forest. I feel like other more

specific statistics could be used.

Response:

Thank you for bringing this to our attention! We have changed the red text to black in Figure 2.

In general, the methods used for marker metabolites screening include the following:

1) Univariate analysis (FC values, P values, FDR values, etc.), 2) Discriminant analysis ((O)PLS-DA method), 3) Machine learning (random forest analysis, support vector machines). Since the data generated by metabolome sequencing are "high-dimensional, high-noise, and high-variance", it is difficult to conduct multiple hypothesis testing when performing univariate analysis. If α correction cannot be applied to the hypothesis test each time, the probability of making one type of error will increase significantly, and the number of false positives increase as well. Therefore, 2) and 3) are generally used to "simplify and downscale" high-dimensional complex data, and they can also retain the maximum amount of original information. However, discriminant analysis cannot deal with nonlinear problems well, and it is easy to overfit, so we need to pay attention to the selection of the principal components. In machine learning, random forest (RF) analysis not only provides an assessment of the importance of variables and helps screen for the most discriminatory metabolites but also explores the best combination of biomarkers to improve the accuracy and stability of classification. Notably, RF significantly outperforms support vector machines in classification problem 6. In addition to this, RF analysis also has the following advantages 7: (1) high processing efficiency; (2) high accuracy; (3) good robustness (good handling of noisy data and missing data); (4)

not easy to overfit.

-Discussion: There is a lot of logic (A and B lead to C and then lead to D) happening with the discussion. This is good to try and follow the authors thought process. However, there is quite a lot of that, in my opinion. I would appreciate a few more citations to help simplify the logic. Also, I think a caveat paragraph would be great to see. Maybe this would be a great place to include my main questions within the manuscript.

Response: Thank you for the comments, we agree! In order to solve your confusion to the maximum extent, we have made the following modifications: we have rewritten 'Discussion' based on your opinion. After this, this manuscript has sought the help of professional English editors to make the article easier to understand.

Reference.

- 1 Allredge, A. L. & Silver, M. W. Characteristics, Dynamics and Significance of Marine Snow. *Prog Oceanogr* **20**, 41-82, doi:Doi 10.1016/0079-6611(88)90053-5 (1988).
- 2 Michaels, A. F. & Silver, M. W. Primary Production, Sinking Fluxes and the Microbial Food Web. *Deep-Sea Res* **35**, 473-490, doi:Doi 10.1016/0198-0149(88)90126-4 (1988).
- 3 Quilliam, M. A., Sim, P. G., McCulloch, A. W. & McInnes, A. G. J. I. J. o. E. A. C. High-performance liquid chromatography of domoic acid, a marine neurotoxin, with application to shellfish and plankton. **36**, 139-154 (1989).
- 4 Sekula-Wood, E. *et al.* Pseudo-nitzschia and domoic acid fluxes in Santa Barbara Basin (CA) from 1993 to 2008. *Harmful Algae* **10**, 567-575, doi:10.1016/j.hal.2011.04.009 (2011).
- 5 Smith, J. *et al.* Persistent domoic acid in marine sediments and benthic infauna along the coast of Southern California. *Harmful Algae* **108**, doi:ARTN 102103 10.1016/j.hal.2021.102103 (2021).
- 6 Gromski, P. S., Xu, Y., Hollywood, K. A., Turner, M. L. & Goodacre, R. J. M. The influence of scaling metabolomics data on model classification accuracy. **11**, 684-695 (2015).
- 7 Liu, Y. J. C. m. & technologies, n. Random forest algorithm in big data environment. **18**, 147-151 (2014).

Response to the Comments of Reviewer #2:

General comments:

-The manuscript attempts to quantify the effect of Domoic Acid (DA) on nitrogen cycle processes in sediments. Earlier work by others showed that diatoms produce a lot of DA, and that it can be transported to sediments by sinking organic material from diatoms growing in surface waters. Thus this project addressed a logical question, so the DA that makes it to ocean sediments affect the biogeochemical processes in those sediments. The experiment was a straightforward incubation design with adequate replication and state of the art analysis of everything from standard water chemistry to metabolomics.

Response: Thank you sincerely for your time reviewing our article. We appreciate the insight you provided and feel it has improved the quality of our paper. In response to some of your comments, we may have poorly presented some information (particularly the effective concentrations of DA in the experimental microcosms). We have explained this and amended the manuscript according to the other points you made. We address each point individually below. Thank you again for your time and consideration!

Specific comments:

- The text needs a good English editor. I was not able to understand some of the text, so I cannot not make a detailed review at this time. In hopes that the presentation and content can both be improved at the next stage, I will provide as much of a review as is possible at a fairly high level, although some of my concerns might be mitigated by

a better understanding of the presentation.

Response: Thank you for your valuable comment! In order to address your confusion to the maximum extent, we have made the following modifications: we have made very detailed modifications about the logic, 'Results', and 'Discussion' of this manuscript, especially the section of 'Results' and 'Discussion'. After this, this manuscript has sought the help of professional English editors whose native language is English.

- The level of Domoic Acid (DA) additions to the microcosms ranged from 0.1mg/L to 1mg/L. A review of the literature, including the two papers cited in this manuscript as the basis for the choice of the DA addition concentration, finds that maximum reported seawater concentrations ranged from 0.220 $\mu\text{g/L}$ (0.00022 mg/L) in the central Pacific (Silver et al. 2010) to $\sim 6 \mu\text{g/L}$ (0.006mg/L) in Santa Barbara Basin (Sekula-Wood et al. 2009). The DA additions to the DA0.5 microcosm exceeds these environmental levels by at least 83-fold.

On the other hand, DA concentration has been measured in sediment traps at up to 163 $\mu\text{g/g}$ dry wt of sediment (Sekula-Wood et al. 2009). If the final concentration of DA in the DA0.5 microcosm was 250 μg in 500 ml (0.5 mg/L in 500 ml, including 200g of sediment), that's 1.25 $\mu\text{g DA/g}$ sediment. By that comparison, the DA additions in this experiment are far below what might be expected to occur in sediments underlying a diatom bloom.

Could the authors please explain their effective DA concentrations? Was the concentration actually measured in the experimental microcosms? The method is cited,

but no measurement data are presented. Does the final concentration (e.g., 0.5 mg/L) refer to the 300 ml of water or the total volume of the microcosm including sediments? If my calculations are correct, it seems unlikely that organisms would ever be subjected to such high bulk water concentrations, but they could experience a large flux of DA to the sediment when bloom debris is deposited. I'm having a hard time figuring out if the experimental additions were high or low. The authors refer to them as trace level, but it's not clear if that is the case.

Response: We appreciate the question and apologize for not clearly explaining the different forms of DA concentrations in the system. In this work, the DA concentrations refer to the concentration of DA in the seawater, i.e., 0.03 mg DA, 0.15 mg DA, and 0.3 mg DA were dissolved in the 300 ml of seawater to form dissolved DA (dDA) at concentrations of 0.1, 0.5, and 1.0 mg/L, respectively. First of all, we should clarify to you the distribution of DA in the marine benthic environment. In the wake of phytoplankton bloom, the DA-ontained algal cells can migrate with the marine snow to the seafloor, and these algal cells usually reach the deep seafloor relatively intact ^{1,2}. After the lysis of these algal cells, DA will be released from the algae cells. Due to the high solubility of DA in seawater ³, most of the DA will be dissolved in the seawater and exist in the dissolved state of DA; only a small fraction of dDA will be adsorbed to the sediment (<10%) ⁴. Therefore, the DA concentration was set based on the concentration of dDA in the environment in this work.

Secondly, the selection of dDA concentrations was based on the data from 15-year continuous fieldwork. It has been confirmed that the concentration of dDA is

commonly about 0.6 mg/L, and the maximum concentration of dDA is about 4.2 mg/L

⁴. Besides, some other evidence can also support the results of this field investigation.

For instance, a field survey in Santa Barbara found that dDA concentration in the marine benthic environment can reach levels of mg/L ⁸. In addition to direct evidence,

a field survey from Washington State found DA concentrations exceeded 0.136 mg/L in surface seawater ⁹. Since the marine benthic environment is actually the ultimate

destination for DA ¹⁰, it is reasonable to speculate that DA concentrations in the

benthic environment can reach mg/L levels. Based on the above information, we

believe the setting of DA concentration in the system is environmentally relevant.

Finally, since another focus of this work is to investigate the biotransformation of DA

in the system (the DA biodegradation time and biodegradation products are provided in Lines 472-474 and Fig. S10, respectively), we monitored the DA concentrations in

seawater and sediments throughout the experiment. The added dDA (0.1, 0.5, and 1

mg/L) could be adsorbed by the sediment, and the maximum concentrations of DA in

the sediments in the three groups were 11.51 ng/g wet sed, 15.75 ng/g wet sed, and 21.11 ng/g wet sed, respectively (the same order of magnitude as the previous field

detection concentrations (levels of ng/g wet sed) ⁵). Most of the DA in the system was

still present in the form of dDA (92.3%, 97.9%, 98.6%), with concentrations of

0.0923, 0.4895, and 0.986 mg/L, respectively. Overall, the exposure concentrations of

DA in the seawater and sediment were both environmentally relevant. We have

redescribed the settings of DA concentration and the DA concentrations in seawater

and sediment in the Introduction and Methods (Lines 49-56; Lines 443-458). Thank

you again for your comments!

- The “abundances” of all the genes are often referred to in the text as “abundances”, not “relative abundances”. This is very misleading. This kind of data is by nature compositional and it’s really hard to translate it to absolute abundances. However, the authors appear to be treating the relative data as absolute abundances, when they refer to changes in abundance of the various N cycle genes (most of page 10, e.g.), etc. The supplemental text does not provide much information on how the abundances were calculated. Were they normalized to total read number only? Or rarified to account for the total number of reads in individual samples? Or better yet, use the centered log transformation (Aitchison 1986). Table S1 provides the relative abundance numbers from which the treatment effects shown in Figure 2 were calculated. But the actual numbers in the Table need to be explained. If it’s simple relative abundance as a fraction of the total reads, these would all be less than 1, or if expressed as a percentage, less than 100. The range of values in Tables S1 and S2 is from zero to >300. We need to know what these numbers are. (I was able to use this table to get the numbers that appear in Figure 2, it’s just not clear where the data in Table S1 came from.)

I assume the gene relative abundance data were also the basis of KEGG pathway analyses, so this “abundance” assumption is also important to acknowledge and deal with here. Correlation analyses are also problematic with proportional data—the Mantel test probably used a Pearson correlation matrix, which is not appropriate. This concern affects all the network analyses as well. It’s possible that a more

rigorous statistical analysis of the compositional relative abundance data would not change the interpretation very much, but the proportional nature of the data must be acknowledged and the implications addressed. I.e., it's a zero sum game. You cannot tell that one gene increased in absolute abundance because all abundance calculations depend on everything else in the sample. Therefore standard correlations don't make sense.

Response: Thank you for your detailed explanation and feedback. We acknowledge that we mistakenly referred to relative abundance as abundance, and did not account for compositional effects ¹¹ in our data analysis. We greatly appreciate your comments and have taken them into serious consideration. We recognize the importance of the issues you raised and will ensure the accuracy and reliability of our data and results in our manuscript. To address the concerns you highlighted, we have made the following modifications:

1. In this work, the abundances of genes were rarified to account for the total number of reads in individual samples, so we have used the term "relative abundance" instead of "abundance" in our manuscript to avoid confusion.
2. According to your suggestion, we used centered log ratio (CLR) to transform the data (including the relative abundance of genes and microorganisms) ¹². The code applied for RStudio is as follows:

```
data <- read.table("data.tsv",header = T,row.names = 1,sep = "\t")
```

```
library(compositions)
```

```
write.table(logratio_data,"clrdata.tsv",sep = "\t")
```

3. After recalculating the data, we remade Figures 4, 5, and 6 based on the recalculated data, and the corresponding results and discussion were also rewritten.

Moreover, we would like to clarify that the Mantel test analysis was based on the Spearman correlation matrix, which was shown in the caption of Figure 4. The relative abundance values greater than 1 are due to the use of copies per million (CoPM) as the unit ¹³, which is a generic analog of the TPM unit used in RNA-seq. We have provided detailed explanations of the data calculation methods as follows:

Regarding gene annotation, we used HUMAnN3 software ¹³ to compare the quality-controlled and host-filtered sequences with the protein database (UniRef90) using DIAMOND. Reads that failed to align were filtered out (using the default parameters of HUMAnN3: translated_query_coverage_threshold= 90.0, prescreen_threshold=0.01, evalue_threshold = 1.0, translated_subject_coverage_threshold = 50.0). The relative abundance (similar to TPM) of each protein in UniRef90 was then calculated. We used the "humann_renorm_table" script provided by HUMAnN to calculate the relative abundance based on the original abundance (with the parameter "--units cpm", which stands for copies per million). We then used the corresponding IDs between UniRef90 and various functional databases (KEGG, MetaCyc, EggNOG, GO, EC, and CAZy) mainly from LinkDB (<https://www.genome.jp/linkdb/>) to calculate the relative abundance of various functional databases.

- Planetary boundary threat? I will need some convincing.

Response: Thank you for your comments! According to the previous paper, if the contaminant is defined as a planetary boundary threat, the following three conditions must be met simultaneously ¹⁴. Condition 1: the contaminant has a disruptive effect on a vital Earth system process. Importantly, we must recognize and accept that we are currently ignorant of this effect, as we are likewise currently ignorant of the planetary boundary; Condition 2: the disruptive effect is not discovered until it is, or inevitably will become, a problem at the planetary scale. If the disruptive effect is discovered before it is a problem at the planetary scale, then it is not a planetary boundary threat since it would be possible to take action to control the pollution before the planetary boundary is exceeded; Condition 3: the effects of the pollution in the environment cannot be readily reversed. If they can be readily reversed, then the exceedance of the planetary boundary could be corrected.

In this work, we defined DA as a potential planetary boundary threat mainly based on the following three aspects: (1) We uncovered for the first time that DA is capable of ecosystem-level effects (DA can act as a stressor to alter the nitrogen cycling in sediments); (2) DA-producing marine harmful algal blooms are pervasive in the global ocean, which means that the impacts of DA inevitably become a problem at the planetary scale; (3) The frequency and intensity of DA-producing harmful algal bloom outbreaks showing an increasing trend in the global ocean, which means that the impact of DA on the ecosystem cannot be readily reversed either.

- Figure 3. We need more information here. What are the units for “counts”? Were the metabolomic data analyzed in triplicate, as was done for the metagenomic analysis?

The methods imply that triplicates were analyzed. If so, can we have some indication of statistical significance of the count data?

Response: Thank you for bringing this to our attention! The “counts” represents the value of Mean Decrease Accuracy, which measures the importance of a metabolite in random forest, so “counts” has no units. Also, the metabolomic data were analyzed in triplicate, as was done for the metagenomic analysis. We have explained the practical meaning of “count” in the caption of Figure 3 (Lines 186-188).

-Table S1. Please state in the caption that these data refer to e.g., DA0.5 day 10. “Treat_10” did not immediately make sense to me.

Response: Thank you for your comments, this has been modified accordingly.

-Figure S2. It would be good to provide units for the rates somewhere on this page. The units are given in the methods, but should be noted here. It seems to me that these data are central to the interpretation of the main point, that actual rates of important N cycle processes were affected by DA. However, these are potential rates (according to the methods, in which very high substrate levels were added to very small incubations). So the significance of DA affects on potential vs in situ rates should be addressed in the text.

These plots are not quantitative enough. I suggest a column plot showing each rate and the error for the three replicate incubations for each experiment. And these data should be in the main text, perhaps as additional plots in Figure 1. The plots should contain information on statistical comparisons between rates among the different treatments and days. The colored dots just don't contain enough information to allow

the reader to evaluate the treatment effects.

Response: Thank you for pointing this out; we agree! The unit for the rates has been added to the Figure. The significance of DA effects on potential vs in situ rates has been addressed in the manuscript (Lines 111-115). In addition, according to your opinion, a column plot was applied to show each rate and the error for the three replicate incubations for each experiment. And the figure is included as the additional plots of Figure 1 in the main text.

-L 233 Summarily—please explain the significance of these particular metabolites.

This statement is too general to mean anything.

Response: We are grateful for this comment. This has been rewritten in lines 180-185, and the significance of these particular metabolites was fully discussed in the “Discussion” (Lines 318-331).

-The metabolite data are interpreted here to show repression of specific pathways – these are central pathways in the metabolism of just about every cell. Does this mean that the DA concentration was so high that all basic metabolism was affected? The phrase “electron provision” is not one with which I am familiar. Does this mean electron transport? Respiration based on redox transfers? I can't make sense of this argument.

Response: Thank you for your comments! First, I need to clarify that the exposure concentrations (0.1, 0.5, and 1.0 mg/L) of DA in this experiment were set based on 15-year continuous field monitoring data, which are environmentally relevant. Secondly, in addition to the AD_{0.5} group being used for metagenomic sequencing,

AD_{0.1} and AD_{1.0} groups were also applied for metagenomic sequencing, with the data available in the NCBI repository (accession number: PRJNA912632). In terms of the N cycling, the three concentrations of DA had the same trend of effect on the relative abundances of genes related to N cycling and electron supply (TCA cycle and glycolysis) of the N cycling. Only the magnitude of the effect was different. We chose the AD_{0.5} group to analyze the mechanism by which DA affects the N cycling because DA concentration is closer to the environmental concentration. Moreover, from our data, DA with concentrations of 0.5 and 1.0 mg/L can affect not only the N cycling and carbon metabolism but also the sulfur cycling and the relative abundance of antibiotic resistance genes, further illustrating the significant environmental impacts of DA. And these findings will be revealed in our future work.

Electron provision refers to the capacity of electron supply, which is a quoted expression from a previous study ¹⁵. To avoid misunderstandings, we have replaced electron provision with electron supply (a more common description) throughout the manuscript. Herein, we take the denitrification process as an example to explain the effect of electron supply on denitrification. During the denitrifying process, the reduction of nitrogen oxide is mainly driven by the electron supply and consumption system ^{16,17}. Electrons are primarily donated as nicotinamide adenine dinucleotide (NADH), which derives from various metabolic pathways of organic substrate, such as the Embden-Meyerhof-Parnas (EMP) pathway, pentose phosphate (PP) pathway and tricarboxylic acid (TCA) cycle during glucose metabolism ¹⁸. Then, the

generated electrons are sequentially consumed by nitrate reductase (NAR), nitrite reductase (NIR), nitric oxide reductase (NOR), and nitrous oxide reductase (NOS), which catalyze the sequential bioreductions of nitrate (NO_3^- -N) to nitrite (NO_2^- -N), nitric oxide (NO), nitrous oxide (N_2O) and N_2 , respectively. Therefore, factors affecting electron supply or electron consumption (i.e. enzyme activity) can affect the denitrification process.

- Figure 4 is very nice and contains a lot of information. The caution about applying simple Spearman correlation analyses to the gene abundance data applies, however. What do the R^2 values refer to in Figure 4b? Each R^2 is adjacent to a box, but each box has a different number of arrows connecting to the box. If the R^2 applies to one of the arrows, that should be stated and made clear in the figure. E.g., for the boxes at the top and bottom of the figure, the R^2 numbers appear to apply to the box, not to a particular arrow. Even if the R^2 did apply to an arrow, what does the R^2 of a path coefficient (r) mean?

Response: Thank you for pointing this out! As mentioned above, the data in Figure 4 has been recalculated, and the figure has also been remade. Moreover, the explanation of the meaning of R^2 has been added to the caption in Figure 4b (Lines 221-222). R^2 represents the proportion of variance explained for every dependent variable (in the box) in the model. Numbers adjacent to the arrows are standardised path coefficients (r), analogous to relative regression weights and indicative of the effect size of the relationship.

-Figures 5 and 6 are perhaps attractive but impossible to read. One cannot see the

taxon id in the left panel of Figure 6. In figure 5, the names of the nodes are unreadable with the low resolution files (worse in the supplemental network figures).

Might be okay reading a high resolution pdf file online.

Response: Thank you for your suggestion! We have modified these two figures to make them to be clearly readable. In addition, we would like to say that we have tried our best to maximize the node names in Figure 5. If you still cannot see them clearly, please enlarge the figure. This figure is a vector image, enlarging the figure will not affect its resolution. Thank you again for your valuable suggestion!

Reference.

- 1 Allredge, A. L. & Silver, M. W. Characteristics, Dynamics and Significance of Marine Snow. *Prog Oceanogr* **20**, 41-82, doi:Doi 10.1016/0079-6611(88)90053-5 (1988).
- 2 Michaels, A. F. & Silver, M. W. Primary Production, Sinking Fluxes and the Microbial Food Web. *Deep-Sea Res* **35**, 473-490, doi:Doi 10.1016/0198-0149(88)90126-4 (1988).
- 3 Quilliam, M. A., Sim, P. G., McCulloch, A. W. & McInnes, A. G. J. I. J. o. E. A. C. High-performance liquid chromatography of domoic acid, a marine neurotoxin, with application to shellfish and plankton. **36**, 139-154 (1989).
- 4 Sekula-Wood, E. *et al.* Pseudo-nitzschia and domoic acid fluxes in Santa Barbara Basin (CA) from 1993 to 2008. *Harmful Algae* **10**, 567-575, doi:10.1016/j.hal.2011.04.009 (2011).
- 5 Smith, J. *et al.* Persistent domoic acid in marine sediments and benthic infauna along the coast of Southern California. *Harmful Algae* **108**, doi:ARTN 102103 10.1016/j.hal.2021.102103 (2021).
- 6 Gromski, P. S., Xu, Y., Hollywood, K. A., Turner, M. L. & Goodacre, R. J. M. The influence of scaling metabolomics data on model classification accuracy. **11**, 684-695 (2015).
- 7 Liu, Y. J. C. m. & technologies, n. Random forest algorithm in big data environment. **18**, 147-151 (2014).
- 8 Umhau, B. P., Benitez-Nelson, C. R., Anderson, C. R., McCabe, K. & Burrell, C. A Time Series of Water Column Distributions and Sinking Particle Flux of Pseudo-Nitzschia and Domoic Acid in the Santa Barbara Basin, California. *Toxins* **10**, doi:ARTN 480 10.3390/toxins10110480 (2018).
- 9 Trainer, V. L. *et al.* Recent domoic acid closures of shellfish harvest areas in Washington State inland waterways. *Harmful Algae* **6**, 449-459, doi:10.1016/j.hal.2006.12.001 (2007).
- 10 Sekula-Wood, E. *et al.* Rapid downward transport of the neurotoxin domoic acid in coastal waters. *Nat Geosci* **2**, 272-275, doi:10.1038/Ngeo472 (2009).
- 11 Gloor, G. B., Wu, J. R., Pawlowsky-Glahn, V. & Egozcue, J. J. It's all relative: analyzing

- microbiome data as compositions. *Ann Epidemiol* **26**, 322-329, doi:10.1016/j.annepidem.2016.03.003 (2016).
- 12 Yuan, M. M. *et al.* Climate warming enhances microbial network complexity and stability. *Nat Clim Change* **11**, 343-U100, doi:10.1038/s41558-021-00989-9 (2021).
- 13 Franzosa, E. A. *et al.* Species-level functional profiling of metagenomes and metatranscriptomes. *Nat Methods* **15**, 962-+, doi:10.1038/s41592-018-0176-y (2018).
- 14 Persson, L. M. *et al.* Confronting Unknown Planetary Boundary Threats from Chemical Pollution. *Environ Sci Technol* **47**, 12619-12622, doi:10.1021/es402501c (2013).
- 15 Jiang, L. *et al.* The metabolic patterns of the complete nitrates removal in the biofilm denitrification systems supported by polymer and water-soluble carbon sources as the electron donors. **342**, 126002 (2021).
- 16 Wan, R., Chen, Y., Zheng, X., Su, Y. & Li, M. Effect of CO₂ on Microbial Denitrification via Inhibiting Electron Transport and Consumption. *Environ Sci Technol* **50**, 9915-9922, doi:10.1021/acs.est.5b05850 (2016).
- 17 Wan, R. *et al.* Tetrabromobisphenol A (TBBPA) inhibits denitrification via regulating carbon metabolism to decrease electron donation and bacterial population. *Water Res* **162**, 190-199, doi:10.1016/j.watres.2019.06.046 (2019).
- 18 Chen, J. & Strous, M. Denitrification and aerobic respiration, hybrid electron transport chains and co-evolution. *Biochim Biophys Acta* **1827**, 136-144, doi:10.1016/j.bbabi.2012.10.002 (2013).

Reviewer #1 (Remarks to the Author):

Over, Li and co-authors addressed most of my comments from before. I really enjoy the overall findings and message of the paper. I also appreciate the adjustments made to the figures and the methods, this really assisted in my understanding.

However, the one difficulty in the paper is the logical long sentences within the results and discussion. This does make it a little harder to read, but it is better than before. Simply shortening or separating out these sentences would greatly improve the manuscript.

Most specific comments:

Line 55: "... a minute amount of DA..." I think the word should be a minimal amount, but I am not sure.

Figure 3: I really like the use of the colormap in this way. However, now that I relook at it, I don't think the red and green combination are a good one. For those red/green color blinded, they will not be able to read it at all.

Line 338: Missing the word 'and'

Reviewer #2 (Remarks to the Author):

The authors responded to my previous comments somewhat thoroughly and appear to have made some of the suggested improvements. It is much more readable and some of the technical issues have been addressed.

One of my major concerns in the first version of this manuscript was about the concentration of DA added to the experiments and whether that concentration was realistic. The authors have confirmed that the concentration added was very high compared to those usually encountered in the water column, but perhaps more realistic compared to those that might occur in a sediment underlying a diatom bloom. This question has been addressed by including references for measured DA concentrations in both water and sediments. L50 – 56.

In the first version of the manuscript, the authors provided the methods for the measurement of DA concentrations but had provided no data on actual measured concentrations. DA concentrations are now reported in the methods section (L453). However, it would have been great to see data on the concentration of DA over time in the incubations, as was done for the DIN concentrations. The concentrations of DA reported for the sediments in the microcosms were on the same order as that previously detected in Santa Barbara Basin sediments. It would have been nice to know when (what day during the incubation) these reported measurements were taken and whether the DA persisted over the time course of the experiment.

How much AOM was added to the incubations? It would be good to know in terms of C or N in order to compare the experimental addition to the level that might be expected in a settling diatom bloom. It appears that AOM concentration might have been measured, since TOC data are used in the correlation analyses (Figure 4a) and the AOM is included in the path analysis (Figure 4b).

I still doubt the planetary boundary effect. The papers cited by the authors documenting the increase in DA due to the increased in HABs and Pseudonitzschia blooms were published up to 3 decades ago. Wouldn't we be seeing an effect by now? How would we know? The authors of the 15 year study cited here attributed the increase in Pseudonitzschia blooms to the Pacific Decadal Oscillation, with the implication that when the PDO relaxes into its opposite phase, the conditions will change back to lower incidences of Pseudonitzschia blooms. So this would be a case of decadal ocean variability, rather than long term climate change. I would argue that we cannot tell either way at this point – the 15 year record does not support long term threat of DA in marine

sediments, but it does not discount the possibility either. Exaggerating the “threat” by conflating decadal natural variability with long term climate change is unnecessary.

Figure 1. In the first review, I suggested including the rate data in the main text, so this is good. The data clearly show an effect of DA on the potential rates, and that the effect is opposite for nitrification vs denitrification and anammox.

I am trying to interpret the DIN data in terms of the potential rates. It would be helpful to make the two sets of data in the same units. I.e., show the DIN data in μM or nM so that the reader can easily compare the concentrations and rates. As a reality check, the reader would ask, how significant is the change in rates compared to the accumulation of DIN? However, since these are potential rates, this comparison cannot make sense. The rates are all potential rates, and thus represent at best the physiological maximum of each process based on the biomass of the active microbes in the sediment. Nitrification rates were measured after 24 h under fully aerated conditions with the additions of 1 mM N. Both denitrification and anammox were measured under fully anoxic conditions with the addition of 100mmol NO_3 (cannot tell what absolute concentration of substrate that might be because the volume of the incubation is not provided) over 2 hours. How can these even be compared to each other? I accept that these rates can show that the presence of DA in the sediments can affect the N cycle processes. I doubt, however, that these measurements provide any insight into the effect of DA in the natural sediment, or that there is any way for the reader to gauge the magnitude of such an effect under in situ conditions.

In L113, it is acknowledged that the potential rates cannot reflect in situ rates, but states that the potential rates do reflect in situ activity. That cannot be true. Potential rates can reflect the potential for activity but cannot tell us that the activity was occurring without stimulation provided by the addition of high substrate levels. Please remove that statement.

I see that the gene levels have been recalculated – the numbers in Figure 2 are slightly different from the last version. I still don't see any quantitative presentation of the degree of significance that can be attached to these numbers and the differences between DA0.5 and AOM or CK. That is concerning because, although many of the gene relative abundances replicated well between replicate treatments, some were much more variable (e.g., *nosZ*, *narG*, *nxrA*). This variability means that even relatively large values for percent difference between treatments might not be significantly different from zero – some error estimate must be provided before you can say that they are significantly higher or lower than some other value.

The gene abundance data are then subjected to correlation analysis and used to infer pathways and relationships between environmental factors and pathways (Fig 4). Some of the results are counterintuitive. For example, how can AOM be positively correlated with nitrification and anammox and denitrification if nitrification is negatively correlated with both denitrification and anammox?

Then some sort of network analysis was performed (Fig 5), but it is not clear to me what data were used in this analysis. Was it the metabolites that represent different KEGG pathways? Where did the phylogenetic assignments of the nodes come from? The methods describing this analysis (L 481, S4 and S5) are very minimal and the text is too cryptic for me to understand.

Description of the modules and their functional significance (Fig. 6) is relatively more clear, except that one cannot figure out where the modules came from or how they might relate to the modules shown in Figure 5.

I don't understand the caption for Figure 6. Where are the red and blue links that imply positive or negative correlations? I don't see any links in this figure.

Line 405 – how would inhibition of denitrification and anammox in the sediment prolong a diatom bloom in surface waters? The sediments used in this experiment were intertidal but most diatom blooms occur somewhat farther offshore. In the SBB, which these authors frequently cite as their model system, the sediments are 500 – 800 m below the surface – hard to imagine a direct feedback between sediment chemistry and surface blooms.

In general the discussion ranges widely and freely over a lot of environmental and ecological material, only loosely tethered to anything quantitative in the data. Causality and consequences are freely attributed to correlations to infer, e.g., direct gene regulation and biogeochemical processes, that are far removed from any actual observations.

Conclusion: The paper provides experimental evidence that DA affects N cycling in sediment microbial assemblages. It does not prove that DA is a danger to the planet or that everything observed in the mesocosms was related directly to DA or could be attributed to DA in the environment. A large number of sophisticated approaches are used to interrogate the experiments, but the descriptions of the methods are so minimal as to be impossible to follow or understand the results.

Response to the Comments of Reviewer #1:

General comments:

Over, Li and co-authors addressed most of my comments from before. I really enjoy the overall findings and message of the paper. I also appreciate the adjustments made to the figures and the methods, this really assisted in my understanding.

However, the one difficulty in the paper is the logical long sentences within the results and discussion. This does make it a little harder to read, but it is better than before.

Simply shortening or separating out these sentences would greatly improve the manuscript.

Response: Thank you very much for your comments. We have now worked on language and readability and have also involved native English speakers for language corrections. We hope that the language level has been substantially improved.

Specific comments:

-Line 55: "... a minute amount of DA..." I think the word should be a minimal amount, but I am not sure.

Response: Thank you for your comment. We agree! "... a minute amount of DA..." has been replaced by "... a minimal amount of DA..."

-Figure 3: I really like the use of the colormap in this way. However, now that I relook at it, I don't think the red and green combination are a good one. For those red/green color blinded, they will not be able to read it at all.

Response: Thank you for bringing this to our attention; you are so detail-oriented! The image's color has been replaced by other colors that are easier to distinguish.

-Line 338: Missing the word 'and'

Response: Thank you for your comments; this has been modified accordingly.

Response to the Comments of Reviewer #2:

General comments:

The authors responded to my previous comments somewhat thoroughly and appear to have made some of the suggested improvements. It is much more readable and some of the technical issues have been addressed.

One of my major concerns in the first version of this manuscript was about the concentration of DA added to the experiments and whether that concentration was realistic. The authors have confirmed that the concentration added was very high compared to those usually encountered in the water column, but perhaps more realistic compared to those that might occur in a sediment underlying a diatom bloom. This question has been addressed by including references for measured DA concentrations in both water and sediments. L50–56.

Response: We are very thankful for your fruitful comments, which will further improve our manuscript. We will revise the manuscript point-by-point according to your comments. Once again, we appreciate your warm work and hope the revision meets the approval requirements.

Specific comments:

-In the first version of the manuscript, the authors provided the methods for the measurement of DA concentrations but had provided no data on actual measured concentrations. DA concentrations are now reported in the methods section (L453). However, it would have been great to see data on the concentration of DA over time in the incubations, as was done for the DIN concentrations. The concentrations of DA

reported for the sediments in the microcosms were on the same order as that previously detected in Santa Barbara Basin sediments. It would have been nice to know when (what day during the incubation) these reported measurements were taken and whether the DA persisted over the time course of the experiment.

Response: Thank you very much for your comments. The data presented in lines 443-451 was measured on the 6th day after adding DA into the microcosm when DA had not yet undergone degradation. DA remains consistently present throughout the experiment (Fig. 1), and we have added a description of the changes in DA concentration in lines 102-113, along with the corresponding figure in the Supplementary Material (Fig. S1). In addition, we also discussed the potential relationship between the concentration of DA and the concentration of DIN in the system (Lines 102-113, aiming to deepen readers' comprehension of the impact of DA on overlying water DIN levels.

Figure 1. Changes of DA concentrations in the system.

-How much AOM was added to the incubations? It would be good to know in terms of C or N in order to compare the experimental addition to the level that might be expected

in a settling diatom bloom. It appears that AOM concentration might have been measured, since TOC data are used in the correlation analyses (Figure 4a) and the AOM is included in the path analysis (Figure 4b).

Response: Thank you for pointing this out. Since both TOC and TON were used in the correlation analysis, we measured the changes in TOC and TOC proportion in different treatments, and the relevant data have been added to the supplementary material (Fig. S7). Initially, we aimed to find data on the content of TOC and TON released from lysed algal cells in marine snow through some field investigations. Unfortunately, we did not find any suitable data for this purpose. However, we determined the quantity of algal cells transported to the sediment by marine snow from a field investigation (described in lines 419-428). Consequently, utilizing the quantified number of these algal cells, we prepared AOM, which was subsequently employed in the experiment.

-I still doubt the-planetary boundary effect. The papers cited by the authors documenting the increase in DA due to the increased in HABs and Pseudonitzschia blooms were published up to 3 decades ago. Wouldn't we be seeing an effect by now? How would we know? The authors of the 15 year study cited here attributed the increase in Pseudonitzschia blooms to the Pacific Decadal Oscillation, with the implication that when the PDO relaxes into its opposite phase, the conditions will change back to lower incidences of Pseudonitzschia blooms. So this would be a case of decadal ocean variability, rather than long term climate change. I would argue that we cannot tell either way at this point—the 15 year record does not support long term threat of DA in marine sediments, but it does not discount the possibility either. Exaggerating the “threat” by

conflating decadal natural variability with long term climate change is unnecessary.

Response: Thank you for pointing this out. As you mentioned, the existing evidence merely suggests that DA may possess planetary boundary effects, but there is no direct evidence to support this conclusion. Hence, we concur with your perspective, and the claim of the planetary boundary effect of DA has been removed in this manuscript.

Thank you again for your feedback.

-I am trying to interpret the DIN data in terms of the potential rates. It would be helpful to make the two sets of data in the same units. I.e., show the DIN data in μM or nM so that the reader can easily compare the concentrations and rates. As a reality check, the reader would ask, how significant is the change in rates compared to the accumulation of DIN? However, since these are potential rates, this comparison cannot make sense. The rates are all potential rates, and thus represent at best the physiological maximum of each process based on the biomass of the active microbes in the sediment. Nitrification rates were measured after 24 h under fully aerated conditions with the additions of 1 mM N. Both denitrification and anammox were measured under fully anoxic conditions with the addition of 100mmol NO_3 (cannot tell what absolute concentration of substrate that might be because the volume of the incubation is not provided) over 2 hours. How can these even be compared to each other? I accept that these rates can show that the presence of DA in the sediments can affect the N cycle processes. I doubt, however, that these measurements provide any insight into the effect of DA in the natural sediment, or that there is any way for the reader to gauge the magnitude of such an effect under in situ conditions.

Response: Thank you for your comment. According to your suggestion, we have presented DIN data in nM units. Moreover, the culture experiment was conducted in executor tubes (8 mL), with a volume of 10 mL of culture medium (This has been added to Text S2). Examining changes in overlying water DIN concentrations and potential nitrogen transformation rates is primarily aimed at investigating the influence of DA on sediment N cycling. These changes can macroscopically reflect DA's influence on sediment N cycling. However, as you mentioned, due to the disparate experimental conditions for these two tests, it becomes challenging to establish a direct correlation between them.

As for in-situ potential rate vs potential rate, we have extensively reviewed literature to determine the extent to which results from cultivation experiments can reflect measurements under in-situ conditions. The findings of the investigation indicate that, although precise numerical values might be elusive, potential rates do indeed offer a certain degree of reflection on the variations of N transformation under in-situ conditions.

Taking the common potential denitrification and in-situ denitrification rates as an example. The study conducted by Findlay, et al. ¹ indicated that potential rates were often comparable to, and broadly overlap, the realized rates. The explanation for roughly equal rates of realized and potential denitrification is that environmental samples perform at peak capacity, and the factors controlling realized denitrification are all close to optimal levels. Furthermore, other studies have indicated consistency between potential rates and in-situ N concentration trends when N cycling functions are

altered. For instance, Zhu et al. confirmed that as the potential N removal rate of sediment decreased (denitrification and Anammox) ², there was an observed increase in the concentrations of overlying water nitrate and nitrite on in-situ sediment. This observation suggests that the potential N transformation rate can, to a certain degree, represent the in-situ rate. Similar findings have also been identified in the investigations of other researchers. Kristensen, E. et al. affirmed that variations in the potential nitrification rate of mangrove sediment paralleled changes in the in-situ observed NO_3^- content within the sediment ³. Furthermore, in the study conducted by Brin, L. D et al., it was discovered that after a thermal disturbance, the potential rate of dissimilatory nitrate reduction to ammonium (DNRA) exhibited a change pattern resembling the in-situ NH_4^+ concentration ⁴.

To sum up, we must acknowledge that potential rates cannot fully represent in-situ rates, primarily due to variations in experimental conditions. Nonetheless, potential nitrogen transformation rates can indeed reflect changes in in-situ N transformation rates to a certain extent, and this method has been widely adopted for studying alterations in in-situ biogeochemical cycling processes ⁵⁻⁷.

-In L113, it is acknowledged that the potential rates cannot reflect in situ rates, but states that the potential rates do reflect in situ activity. That cannot be true. Potential rates can reflect the potential for activity but cannot tell us that the activity was occurring without stimulation provided by the addition of high substrate levels. Please remove that statement.

Response: Thank you for your valuable comment! We have removed this statement.

-I see that the gene levels have been recalculated—the numbers in Figure 2 are slightly different from the last version. I still don't see any quantitative presentation of the degree of significance that can be attached to these numbers and the differences between DA0.5 and AOM or CK. That is concerning because, although many of the gene relative abundances replicated well between replicate treatments, some were much more variable (e.g., *nosZ*, *narG*, *nxA*). This variability means that even relatively large values for percent difference between treatments might not be significantly different from zero – some error estimate must be provided before you can say that they are significantly higher or lower than some other value.

Response: Thank you for bringing this to our attention! First of all, we would like to state that the changes in the relative abundance of genes in Fig. 2 are calculated based on the "rarified" gene data, not on the "centered log transformation" gene data. The reason is that some genes relative abundance data will become negative after "centered log transformation", making it impossible to calculate percentage changes. Apart from this, the gene data used for Fig. 4, 5, and 6 are all "centered log transformation" gene data. As for the slight modification, it is due to the fact that we miscalculated the relative abundance change values of the genes *ureC* and *hzsB*, so we modified the change values of the relative abundance of these two genes in the previous revised manuscript.

Secondly, according to your suggestion, we conducted a significance analysis of the percentage difference between treatments via Tukey's test. We have marked the percentages with significant differences on the figure using asterisks (*) and have revised the corresponding descriptions for Figure 2 and Fig. S2 (Lines 133-151). Thank

you again for your valuable suggestion!

-The gene abundance data are then subjected to correlation analysis and used to infer pathways and relationships between environmental factors and pathways (Fig 4). Some of the results are counterintuitive. For example, how can AOM be positively correlated with nitrification and anammox and denitrification if nitrification is negatively correlated with both denitrification and anammox?

Response: Thank you for your comment. In this work, Fig. 4b showed that nitrification can positively impact Anammox, but it has no impact on denitrification. However, we surmise that your intended inquiry revolves around this question: if nitrification is positively correlated with Anammox, then how can DA be positively correlated with nitrification and negatively correlated with Anammox? Evidently, the results mentioned above may seem counterintuitive, but they can be elucidated through a comprehension of the fundamental modeling principles of PLS-PM. Firstly, it is imperative to acknowledge that our statement might have led to the misconception; here, positive correlation and negative correlation refer to positive and negative impacts (This has been modified on line 216). In principle, PLS-PM is a fundamental model that can identify and quantify the overall influence of an environmental factor on ecological processes within a system (direct impact, indirect impact). Taking DA's impact on the Anammox process as an example, the path coefficient (-0.51) reflects the total impact of DA on the Anammox process, encompassing both direct and indirect impacts. Herein, the indirect impact of DA refers to its indirect impact on Anammox by influencing other environmental factors or nitrogen cycling processes within the system. For instance,

DA could impact nitrification, and nitrification, in turn, affects Anammox, resulting in DA's indirect influence on Anammox (Fig. 4b). Despite nitrification having a positive influence on Anammox, the sum of the direct and indirect effects yields a negative value, leading to a negative impact of DA on Anammox. To sum up, the interplay of positive and negative relationships within the path model can create intricate effects; at times, one path's impact might partially counteract another path's effect, yielding outcomes that might not be intuitive.

If this explanation remains unclear, let us provide a simpler example to enhance your understanding of path effects. Please note that this example differs in nature from the analytical process described above; it merely illustrates that counterintuitive paths aren't necessarily incorrect.

Consider the relationship between an individual's income, expenditures, and savings:

- The "Income -> Savings" path has a positive impact: If income increases, savings might also increase due to more disposable funds.
- The "Income -> Expenditures" path has a positive impact: If income increases, expenditures might also rise, as individuals might spend more.
- The "Expenditures -> Savings" path has a negative impact: If expenditures rise, savings might decrease, as less money remains.

However, what's intriguing here is that despite the positive correlation between income and expenditures and the negative correlation between expenditures and savings, the relationship between income and savings remains positive. This could be because even though increased expenditures might decrease savings, a substantial increase in income

could still lead to a net growth in savings. We hope our explanation can solve your confusion.

-Then some sort of network analysis was performed (Fig 5), but it is not clear to me what data were used in this analysis. Was it the metabolites that represent different KEGG pathways? Where did the phylogenetic assignments of the nodes come from? The methods describing this analysis (L 481, S4 and S5) are very minimal and the text is too cryptic for me to understand.

Response: Thank you for your comment. Fig. 5 illustrated the impact of endogenous dynamics during microbial community assembly on the N cycling process. The data used for Fig. 5 correspondingly represent microbial taxa (at the family level) and various N cycling pathways. This work determined the activity of different N cycling pathways through KEGG module analysis. KEGG module analysis is a metagenomic analysis method that examines the activity of various biochemical processes at the pathway level. Furthermore, we have revised the relevant analytical methods to enhance their comprehensibility and detail (Lines 472-487, Text S4, Text S5). Once again, we appreciate your valuable suggestions!

-Description of the modules and their functional significance (Fig. 6) is relatively more clear, except that one cannot figure out where the modules came from or how they might relate to the modules shown in Figure 5.

Response: We appreciate your feedback. The steps for module analysis are as follows:

- 1) N cycling genes and microorganisms (genus level, relative abundances exceeding 0.05%) were utilized for constructing the co-occurrence network. The co-

occurrence network analysis was conducted using the "Hsmic" package.

2) Network characteristics were calculated using the "igraph" package, and the network was visualized using Gephi (v0.9.6) and Cytoscape (v3.9.1). Furthermore, the generated co-occurrence network was modularized and visualized using the MCODE plugin within Cytoscape. This has been redescribed in "Methods" (Lines 503-510).

As stated in the manuscript (Lines 234-236), Figures 5 and 6 were applied to elucidate the mechanism by which DA alters N cycling from the microbial ecological perspective. Firstly, a co-occurrence network analysis was conducted between microorganisms (at the family level) and N cycling pathways (Fig. 5), aiming to illustrate the impact of biotic endogenous dynamics generated during microbial community assembly on N cycling processes. Hence, we did not perform the modular analysis of Fig. 5. Additionally, considering the microbial ecology standpoint that changes in microbial function are attributed to shifts in ecological niches (considered as the module in the co-occurrence network, lines 271-272), we performed modular analysis on the co-occurrence network of microorganisms (at the genus level) and N cycling genes (Fig. 6), aiming to elucidate the impact mechanism of DA on N cycling from the microbial ecological perspective.

-I don't understand the caption for Figure 6. Where are the red and blue links that imply positive or negative correlations? I don't see any links in this figure.

Response: Thank you for your comments. The links in Fig. 6 are embedded between points within the module, i.e., between N-cycling genes and genus-level microbes within the same module. Due to the dense interconnections between nodes within the

module, they almost saturate the entire module, making them challenging to distinguish. We acknowledge that these links are difficult for readers to discern. However, the excessive number of links generated within the module makes their separation impractical. We explain these links in the figure caption to address this issue, hoping to alleviate any confusion (Lines 291-293). Thank you again for your feedback.

-Line 405-how would inhibition of denitrification and anammox in the sediment prolong a diatom bloom in surface waters? The sediments used in this experiment were intertidal but most diatom blooms occur somewhat farther offshore. In the SBB, which these authors frequently cite as their model system, the sediments are 500-800 m below the surface-hard to imagine a direct feedback between sediment chemistry and surface blooms.

Response: We sincerely appreciate your feedback. In light of existing literature highlighting the significant influence of offshore sediments on dissolved nutrient concentrations and fluxes in the marine environment ⁸, we put forth this statement. However, as you rightly pointed out, most diatom blooms happen farther offshore, with deeper water. This weakens the link between sediment and surface seawater, lessening the effect of sediment-related changes on surface water nutrient concentrations. We indeed overlooked this aspect, and as a result, this statement has been removed from the manuscript.

-In general, the discussion ranges widely and freely over a lot of environmental and ecological material, only loosely tethered to anything quantitative in the data. Causality and consequences are freely attributed to correlations to infer, e.g., direct gene

regulation and biogeochemical processes, that are far removed from any actual observations.

Response: Thank you for your comment. We have modified the discussion to be toned down, speculation reduced (including removing claims regarding planetary boundary level threat, direct gene regulation and biogeochemical processes, etc.), and focused on what our results can actually support (Lines 346-350; 390-394;404-411).

-Conclusion: The paper provides experimental evidence that DA affects N cycling in sediment microbial assemblages. It does not prove that DA is a danger to the planet or that everything observed in the mesocosms was related directly to DA or could be attributed to DA in the environment. A large number of sophisticated approaches are used to interrogate the experiments, but the descriptions of the methods are so minimal as to be impossible to follow or understand the results.

Response: Thank you for taking the time to review our manuscript. We have revised the manuscript point-by-point following your comments, especially in the Methods and the Discussion. Once again, we appreciate your warm work earnestly and hope the revision will meet the approval requirements.

- 1 Findlay, S. E. G. *et al.* Cross-stream comparison of substrate-specific denitrification potential. *Biogeochemistry* **104**, 381-392, doi:10.1007/s10533-010-9512-8 (2011).
- 2 Zhu, L. *et al.* Algal Accumulation Decreases Sediment Nitrogen Removal by Uncoupling Nitrification-Denitrification in Shallow Eutrophic Lakes. *Environ Sci Technol* **54**, 6194-6201, doi:10.1021/acs.est.9b05549 (2020).
- 3 Kristensen, E. *et al.* Transformation and transport of inorganic nitrogen in sediments of a southeast Asian mangrove forest. *Aquat Microb Ecol* **15**, 165-175, doi:DOI 10.3354/ame015165 (1998).
- 4 Brin, L. D., Giblin, A. E. & Rich, J. J. Effects of experimental warming and carbon addition on nitrate reduction and respiration in coastal sediments. *Biogeochemistry* **125**, 81-95, doi:10.1007/s10533-015-0113-4 (2015).
- 5 Shi, W. Q. *et al.* Wind induced algal migration manipulates sediment denitrification N-loss

- patterns in shallow Taihu Lake, China. *Water Res* **209**, doi:ARTN 117887
10.1016/j.watres.2021.117887 (2022).
- 6 Jiang, X. Y. *et al.* Role of algal accumulations on the partitioning between N₂ production and dissimilatory nitrate reduction to ammonium in eutrophic lakes. *Water Res* **183**, doi:ARTN 116075
10.1016/j.watres.2020.116075 (2020).
- 7 Seeley, M. E., Song, B., Passie, R. & Hale, R. C. Microplastics affect sedimentary microbial communities and nitrogen cycling. *Nature Communications* **11**, doi:10.1038/s41467-020-16235-3 (2020).
- 8 Murray, L. G., Mudge, S. M., Newton, A. & Icely, J. D. The effect of benthic sediments on dissolved nutrient concentrations and fluxes. *Biogeochemistry* **81**, 159-178, doi:10.1007/s10533-006-9034-6 (2006).

Reviewer #2 (Remarks to the Author):

The authors have addressed most of my suggestions and concerns adequately and I think their revisions have improved the rigor, as well as the clarity and readability, of the manuscript. They have also acknowledged the limitations of the work appropriately, which in no way diminishes the vast quantity and high quality of the work presented.

A few minor comments:

It's great that the domoic acid measurements are included in the supplemental figures. However, the DA values cannot truly be said to "fluctuate", as stated in Line 103. It would be more accurate and informative to say that "DA concentrations declined gradually after Day 9 in all three treatments." To say "fluctuate" actually does an injustice to the nice pattern of decrease that your data provide.

L103 The phrase "a certain amount of association with DIN concentrations" is too vague to be useful. Was there a significant correlation? Or just leave out this phrase, as the relationship is explicitly described below.

Figure 3. The heat map does not show a wide enough range of color. So we can't really see the difference in metabolite level in the plot. The scale on the heat map appears to be fold difference? Please define the scale and increase the color range.

Figure 5. The nodes are in different sizes. What is the significance of node size? Include this information in the caption.

Response to the Comments of Reviewer #2:

It's great that the domoic acid measurements are included in the supplemental figures. However, the DA values cannot truly be said to "fluctuate", as stated in Line 103. It would be more accurate and informative to say that "DA concentrations declined gradually after Day 9 in all three treatments" To say "fluctuate" actually does an injustice to the nice pattern of decrease that your data provide.

Response: Thank you very much for your comments. We have changed the sentence on line 103 to "DA concentrations declined gradually after Day 9 in all three treatments" (Lines 104-105).

Specific comments: L103 The phrase "a certain amount of association with DIN concentrations" is too vague to be useful. Was there a significant correlation? Or just leave out this phrase, as the relationship is explicitly described below.

Response: Thank you very much for your comments. We have leaved out this phrase.

Specific comments: Figure 3. The heat map does not show a wide enough range of color. So we can't really see the difference in metabolite level in the plot. The scale on the heat map appears to be fold difference? Please define the scale and increase the color range.

Response: Thank you for your valuable comment! We have defined the scale (Lines 766-768) and increased the color range of Figure 3.

Specific comments: Figure 5. The nodes are in different sizes. What is the significance of node size? Include this information in the caption.

Response: Thank you for your valuable comment! We have included the nodes

information in the caption of Figure 5 (Lines 787-789).